# EFFICIENTLY COMPUTING SIMILARITIES TO PRIVATE DATASETS

**Arturs Backurs**
Microsoft Research
arturs.backurs@gmail.com

**Zinan Lin**
Microsoft Research
zinanlin@microsoft.com

**Sepideh Mahabadi**
Microsoft Research
mahabadi@ttic.edu

**Sandeep Silwal**
MIT
silwal@mit.edu

**Jakub Tarnawski**
Microsoft Research
jakub.tarnawski@microsoft.com

## ABSTRACT

Many methods in differentially private model training rely on computing the similarity between a query point (such as public or synthetic data) and private data. We abstract out this common subroutine and study the following fundamental algorithmic problem: Given a similarity function $f$ and a large high-dimensional private dataset $X \subset \mathbb{R}^d$, output a differentially private (DP) data structure which approximates $\sum_{x \in X} f(x, y)$ for any query $y$. We consider the cases where $f$ is a kernel function, such as $f(x, y) = e^{-\|x-y\|_2^2/\sigma^2}$ (also known as DP kernel density estimation), or a distance function such as $f(x, y) = \|x - y\|_2$, among others.

Our theoretical results improve upon prior work and give better privacy-utility trade-offs as well as faster query times for a wide range of kernels and distance functions. The unifying approach behind our results is leveraging 'low-dimensional structures' present in the specific functions $f$ that we study, using tools such as provable dimensionality reduction, approximation theory, and one-dimensional decomposition of the functions. Our algorithms empirically exhibit improved query times and accuracy over prior state of the art. We also present an application to DP classification. Our experiments demonstrate that the simple methodology of classifying based on average similarity is orders of magnitude faster than prior DP-SGD based approaches for comparable accuracy.

## 1 INTRODUCTION

It is evident that privacy is an important and often non-negotiable requirement in machine learning pipelines. In response, the rigorous framework of differential privacy (DP) has been adopted as the de-facto standard for understanding and alleviating privacy concerns (Dwork et al., 2006). This is increasingly relevant as non-private ML models have been shown to profusely leak sensitive user information (Fredrikson et al., 2015; Carlini et al., 2019; Chen et al., 2020a; Carlini et al., 2021; 2022; 2023a; Haim et al., 2022; Tramèr et al., 2022; Carlini et al., 2023b). Many methodologies have been proposed in hopes of balancing DP requirements with retaining good downstream performance of ML models. Examples include generating public synthetic data *closely resembling* the private dataset at hand (and training on it) (Lin et al., 2020; Li et al., 2021; Yu et al., 2021; Yin et al., 2022; Yue et al., 2022; Lin et al., 2023), or selecting *similar* public examples for pre-training ML models (Hou et al., 2023; Yu et al., 2023). Furthermore, in the popular DP-SGD method, it is also widely understood that the use of public data bearing *similarity* to private datasets vastly improves downstream performance (Yu et al., 2020; 2021; Yin et al., 2022; Li et al., 2021; De et al., 2022). To list some concrete examples, Hou et al. (2023) use a variant of the Fréchet distance to compute similarities of private and public data, Yu et al. (2023) use a trained ML model to compute the similarity between public and private data, and Lin et al. (2023) use a voting scheme based on the $\ell_2$ distances between (embeddings of) private and synthetic data to select synthetic representatives.

Common among such works is the need to *compute similarities to a private dataset*. While this is explicit in examples such as (Hou et al., 2023; Yu et al., 2023; Lin et al., 2023), it is also implicit in

many other works which employ the inductive bias that pre-training on *similar* public data leads to better DP model performance (Yu et al., 2020; Li et al., 2021; De et al., 2022; Yin et al., 2022).

We study the abstraction of this key subroutine and consider the following fundamental algorithmic problem: given a private dataset $X \subset \mathbb{R}^d$ and a similarity function $f(x, y) : \mathbb{R}^d \times \mathbb{R}^d \to \mathbb{R}$, such as a kernel or distance function, output a *private* data structure $\mathcal{D}_X : \mathbb{R}^d \to \mathbb{R}$ which approximates the map $y \to \sum_{x \in X} f(x, y)$. We additionally require that $\mathcal{D}_X$ be always private with respect to $X$, regardless of the number of times it is queried. This problem has garnered much recent interest due to its strong motivation (Hall et al., 2013; Huang & Roth, 2014; Wang et al., 2016; Aldà & Rubinstein, 2017; Coleman & Shrivastava, 2021; Wagner et al., 2023). It is a meaningful abstraction since similarities between (neural network based) embeddings can meaningfully represent rich relationships between objects such as images or text (Radford et al., 2021). Indeed, beyond training private models, there is additional motivation from performing downstream tasks such as private classification or clustering, where a natural methodology is to classify a query as the class which it has the highest similarity to.

In addition to the privacy angle, computing similarity to a dataset is a fundamental and well-studied problem in its own right. In the case where $f$ is a kernel such as $f(x, y) = e^{-\|x-y\|_2/\sigma^2}$, this is known as kernel density estimation (KDE), whose non-private setting has been extensively studied (Backurs et al., 2018; 2019; Charikar et al., 2020; Bakshi et al., 2022), with many applications in machine learning; see (Schölkopf et al., 2002; Shawe-Taylor et al., 2004; Hofmann et al., 2008) for a comprehensive overview. In the case where $f$ is a distance function, the sum represents the objective of various clustering formulations, such as $k$-means and $k$-median.

## 1.1 OUR RESULTS

The aforementioned works have produced non-trivial utility-privacy trade-offs for computing similarities privately for a wide class of $f$. On the theoretical side, we improve upon these results by giving faster algorithms and improved utility-privacy trade-offs for a wide range of kernels and distance functions. We also study utility lower bounds in order to understand the inherent algorithmic limitations for such problems. Our algorithms are also validated practically; they demonstrate empirical improvements over baselines, both in terms of accuracy and query time.

**Definitions.** We wish to design differentially private algorithms (see Definition A.1 for a DP primer) under the following natural and standard notion: datasets $X, X'$ are neighboring if they differ on a single data point. We also work under the function release model; the data structure we output must handle arbitrarily many queries without privacy loss.

**Function release.** Given a private dataset $X$ and a public function $f$, we wish to release a differentially private (DP) data structure capable of answering either kernel density estimation (KDE) or distance queries. We focus on the function release model as in Wagner et al. (2023) and employ their definition: the algorithm designer releases a description of a data structure $\mathcal{D}$ which itself is private (i.e. $\mathcal{D}$ is the output of a private mechanism as per Definition A.1). A client can later use $\mathcal{D}$ to compute $\mathcal{D}(y)$ for any query $y$. Since $\mathcal{D}$ itself satisfies $\varepsilon$-DP, it can support an *arbitrary number* of queries without privacy loss. This is motivated by scenarios such as synthetic data generation, or when we do not have a pre-specified number of queries known upfront. Our accuracy guarantees are also stated similarly as in Wagner et al. (2023): we bound the error for any fixed query $y$. Thus, while our outputs are always private (since $\mathcal{D}$ itself is private), some query outputs can be inaccurate.

Our private dataset is denoted as $X \subset \mathbb{R}^d$, with $|X| = n$. The similarities are computed with respect to a public function $f : \mathbb{R}^d \times \mathbb{R}^d \to \mathbb{R}$. We define distance queries (for distance functions such as $\|x - y\|_2$) and KDE queries (for kernel functions such as $f(x, y) = e^{-\|x-y\|_2^2}$) as follows:

**Definition 1.1** (Distance Query). *Let $f$ be a distance function. Given a query $y \in \mathbb{R}^d$, a distance query computes an approximation to $\sum_{x \in X} f(x, y)$.*

**Definition 1.2** (KDE Query). *Let $f$ be a kernel function. Given a query $y \in \mathbb{R}^d$, a KDE query computes an approximation to $\frac{1}{|X|} \sum_{x \in X} f(x, y)$.*

The normalization by $|X| = n$ is inconsequential; we follow prior convention for KDE queries. For distance queries, the un-normalized version seemed more natural to us.

| Type | $f$ | Thm. | Our Error | Prior Error | Our Query Time | Prior Query Time |
|---|---|---|---|---|---|---|
| Distance Queries | $\|x-y\|_1$ | B.3 | $\left(1+\alpha, \frac{d^{1.5}}{\varepsilon\sqrt{\alpha}}\right)$ | $\left(1, \left(\frac{nd^7}{\varepsilon^2}\right)^{1/3}\right)$ | $d$ | $d$ |
| | $\|x-y\|_2$ | D.2 | $\left(1+\alpha, \frac{1}{\varepsilon\alpha^{1.5}}\right)$ | $\left(1, \left(\frac{n^{15}d^7}{\varepsilon^2}\right)^{1/17}\right)$ | $d$ | $\geq d$ |
| | $\|x-y\|_2^2$ | D.5 | $\left(1, \frac{d}{\varepsilon}\right)$ | - | $d$ | - |
| | $\|x-y\|_p^p$ | D.4 | $\left(1+\alpha, \frac{d}{\varepsilon\sqrt{\alpha}}\right)$ | - | $d$ | - |
| KDE Queries | $e^{-\|x-y\|_2}$ | E.1 | $(1,\alpha)$ | $(1,\alpha)$ | $d+\frac{1}{\alpha^4}$ | $\frac{d}{\alpha^2}$ |
| | $e^{-\|x-y\|_2^2}$ | E.1 | $(1,\alpha)$ | $(1,\alpha)$ | $d+\frac{1}{\alpha^4}$ | $\frac{d}{\alpha^2}$ |
| | $\frac{1}{1+\|x-y\|_2}$ | F.1 | $(1,\alpha)$ | - | $d+\frac{1}{\alpha^4}$ | - |
| | $\frac{1}{1+\|x-y\|_2^2}$ | F.1 | $(1,\alpha)$ | $(1,\alpha)$ | $d+\frac{1}{\alpha^4}$ | $\frac{d}{\alpha^2}$ |
| | $\frac{1}{1+\|x-y\|_1}$ | F.1 | $(1,\alpha)$ | - | $\frac{d}{\alpha^2}$ | - |

Table 1: Summary of the $\varepsilon$-DP upper bounds. See Definition 1.3 for the error notation. For clarity, we suppress all logarithmic factors. The KDE bounds assume that $n \geq \tilde{\Omega}\left(\frac{1}{\alpha\varepsilon^2}\right)$. The distance query bounds are stated for points in a bounded radius. **The prior distance query results are due to Huang & Roth (2014), and the prior KDE results are due to Wagner et al. (2023).**

**Discussion of Theoretical Results.** The main trade-off that we are interested in is between privacy, as measured with respect to the standard DP definition (Definition A.1), and accuracy of our answers, also called utility. For example, a data structure which always returns a fixed answer, such as 42, is clearly always private regardless of the number of queries performed, but is highly inaccurate. Thus, our goal is to obtain non-trivial accuracy guarantees while respecting privacy. Secondary, but important, concerns are query time and data structure construction time and space. Our main theoretical results are summarized in Table 1, where we use the following error notation.

**Definition 1.3** (Error Notation). *For a fixed query, if $Z$ represents the value output by our private data structure and $Z'$ represents the true value, we say that $Z$ has error $(M, A)$ for $M \geq 1$ and $A \geq 0$ if $\mathbb{E}[|Z - Z'|] \leq (M-1)Z' + A$. That is, we have relative error $M-1$ and additive error $A$. The expectation is over the randomness used by our data structure.*

We want $M$ to be close to $1$ and the additive error $A$ to be as small as possible. Table 1 shows our errors and query times, as well as those of the most relevant prior works. See Section 2 for a technical overview of how we obtain these results.

For distance queries, the most relevant work is Huang & Roth (2014). They considered the $\ell_1$ and $\ell_2$ functions and obtained additive errors with large dependence on $n$ (dataset size) and $d$ (dimension); see Table 1. In contrast, we show that if we allow for a small multiplicative error (e.g. $\alpha = 0.001$ in Table 1), we can obtain additive error with improved dependence on $d$ and *no dependence on $n$*.

**Theorem 1.1** (Informal; see Theorem B.3 and Corollary D.2). *Suppose the data points have bounded diameter in $\ell_1$. For any $\alpha \in (0,1)$ and $\varepsilon > 0$, there exists an algorithm which outputs an $\varepsilon$-DP data structure $\mathcal{D}$ capable of answering any $\ell_1$ distance query with $\left(1+\alpha, \tilde{O}\left(\frac{d^{1.5}}{\varepsilon\sqrt{\alpha}}\right)\right)$ error.*

*For the $\ell_2$ case, where the points have bounded $\ell_2$ diameter, we obtain error $\left(1+\alpha, \tilde{O}\left(\frac{1}{\varepsilon\alpha^{1.5}}\right)\right)$.*

Our approach is fundamentally different, and much simpler, than that of Huang & Roth (2014), who used powerful black-box online learning results to approximate the sum of distances. Furthermore, given that we think of $n$ as the largest parameter, we incur much smaller additive error. Our simpler approach also demonstrates superior empirical performance as discussed shortly. Our $\ell_1$ upper bounds are complemented with a lower bound stating that any $\varepsilon$-DP algorithm supporting $\ell_1$ distance queries for private datasets in the box $[0, R]^d$ must incur $\tilde{\Omega}(Rd/\varepsilon)$ error.

**Theorem 1.2** (Informal; see Theorem C.2). *Any $\varepsilon$-DP data structure which answers $\ell_1$ distance queries with additive error at most $T$ for any query must satisfy $T = \tilde{\Omega}(Rd/\varepsilon)$.*

Note that our lower bound only pertains to additive error and does not say anything about multiplicative error. It is an interesting direction to determine if multiplicative factors are also necessary. We also obtain results for other functions related to distances, such as $\ell_p^p$; see Appendix D.

We now discuss our kernel results. For the Gaussian ($e^{-\|x-y\|_2^2}$), exponential ($e^{-\|x-y\|_2}$), and Cauchy ($\frac{1}{1+\|x-y\|_2^2}$) kernels, we parameterize our runtimes in terms of additive error $\alpha$. Here, we obtain query times of $\tilde{O}(d + 1/\alpha^4)$ whereas prior work (Wagner et al., 2023) requires $\tilde{O}(d/\alpha^2)$ query time. Thus our results are faster in high-dimensional regimes where $d \gg 1/\alpha^2$.

**Theorem 1.3** (Informal; see Theorems E.1 and F.1). *Consider the Gaussian, exponential, and Cauchy kernels. In each case, for any $\varepsilon > 0$ and $\alpha \in (0, 1)$, there exists an algorithm which outputs an $\varepsilon$-DP data structure that answers KDE queries with error $(1, \alpha)$ and query times $\tilde{O}(d + 1/\alpha^4)$.*

For kernels $\frac{1}{1+\|x-y\|_2}$ and $\frac{1}{1+\|x-y\|_1}$, we obtain the first private data structures; see Table 1. We do this via a black-box reduction to other kernels that already have private data structure constructions, using tools from function approximation theory; this is elaborated more in Section 2. All KDE results, including prior work, assume that $n$ is lower-bounded by some function of $\alpha$ and $\varepsilon$. These two kernels and the Cauchy kernel fall under the family of *smooth* kernels (Backurs et al., 2018).

We also give faster query times for the *non-private* setting for the Gaussian, exponential, and Cauchy KDEs. Interestingly, our improvements for the non-private setting use tools designed for our private data structures and are faster in the large $d$ regime.

**Theorem 1.4** (Informal; see Theorem G.1). *For the Gaussian kernel, we improve prior running time for computing a non-private KDE query with additive error $\alpha$ from $\tilde{O}\left(\frac{d}{\varepsilon^2 \alpha^{0.173+o(1)}}\right)$ to $\tilde{O}\left(d + \frac{1}{\varepsilon^2 \alpha^{2.173+o(1)}}\right)$. Similarly for the exponential kernel, the improvement in the query time is from $\tilde{O}\left(\frac{d}{\varepsilon^2 \alpha^{0.1+o(1)}}\right)$ to $\tilde{O}\left(d + \frac{1}{\varepsilon^2 \alpha^{2.1+o(1)}}\right)$. The preprocessing time of both algorithms is asymptotically the same as in prior works.*

**Discussion of Empirical Results.** Our experimental results are given in Section 4. We consider three experiments which are representative of our main results. The first setting demonstrates that our $\ell_1$ query algorithm is superior to prior state of the art (Huang & Roth, 2014) for accurately answering distance queries. The error of our algorithm smoothly decreases as $\varepsilon$ increases, but their algorithms always return the trivial estimate of 0. This is due to the fact that the constants used in their theorem are too large to be practically useful. We also demonstrate that our novel dimensionality reduction results can be applied black-box in conjunction with any prior DP-KDE algorithm, leading to savings in both data structure construction time and query time, while introducing negligible additional error.

Lastly, we explore an application to DP classification on the CIFAR-10 dataset. The standard setup is to train a private classification model on the training split (viewed as the private dataset), with the goal of accurately classifying the test split (Yu et al., 2020; De et al., 2022). Our methodology is simple, fast, and does not require a GPU: we simply instantiate a private similarity data structure for each class and assign any query to the class which it has the highest similarity to (or smallest distance if $f$ is a distance). We set $f$ to be $\ell_2^2$ since it has arguably the simplest algorithm. In contrast to prior works, our methodology involves no DP-SGD training. For comparable accuracy, we use $>$ **3 orders of magnitude** less runtime compared to prior baselines (Yu et al., 2020; De et al., 2022).

## 2 TECHNICAL OVERVIEW

At a high level, all of our upper bounds crucially exploit fundamental 'low-dimensionality structures' present in the $f$'s that we consider. For different $f$'s, we exploit different 'low-dimensional' properties, elaborated below, which are tailored to the specific $f$ at hand. However, we emphasize that the viewpoint of 'low-dimensionality' is *the* extremely versatile tool driving all of our algorithmic work. We provide the following insights into the low-dimensional properties used in our upper bounds.

**Distance Queries via One-dimensional Decompositions.** For the $\ell_1$ distance function, our improvements are obtained by reducing to the one-dimensional case. To be more precise, we use the well-known property that $\sum_{x \in X} \|x - y\|_1 = \sum_{j=1}^{d} \sum_{x \in X} |x_j - y_j|$. In other words, the sum of $\ell_1$ distances decomposes into $d$ one-dimensional sums (this is also true for other related functions such as $\ell_p^p$). This explicit low-dimensional representation offers a concrete avenue for algorithm design: we create a differentially private data structure for each of the $d$ one-dimensional projections of the dataset. In one dimension, we can employ many classic and efficient data structure tools. Furthermore, using metric embedding theory (Theorem D.1), we can also embed $\ell_2$ into $\ell_1$ using an *oblivious* map, meaning that any algorithmic result for $\ell_1$ implies similar results for $\ell_2$ as well.

**Kernels via New Dimensionality Reduction Results.** For kernels such as Gaussian, exponential, and Cauchy, we obtain novel dimensionality reduction results. Our results show that KDE values are preserved if we project both the dataset and the queries to a suitably low dimension via an *oblivious*, data-independent linear map. Our dimensionality reduction schemes are automatically privacy-respecting: releasing an oblivious, data-independent matrix leaks no privacy. Our results also have implications for *non-private* KDE queries and give new state-of-the-art query times.

To obtain our new dimensionality reduction bounds, we analyze Johnson-Lindenstrauss (JL) matrices for preserving sums of kernel values. The main challenge is that kernel functions are *non-linear* functions of distances, and preserving distances (as JL guarantees) does not necessarily imply that non-linear functions of them are preserved. Furthermore, JL-style guarantees may not even be *true*. JL guarantees that distances are preserved up to relative error when projecting to approximately $O(\log n)$ dimensions (Johnson & Lindenstrauss, 1984), but this is not possible for the kernel values: if $\|x - y\|_2$ is extremely large, then after applying a JL projection $G$, $\|Gx - Gy\|_2$ can differ from $\|x - y\|_2$ by a large *additive* factor $\Delta$ (even if the relative error is small) with constant probability, and thus $e^{-\|Gx - Gy\|_2} = e^{-\Delta} \cdot e^{-\|x - y\|_2}$ *does not* approximate $e^{-\|x - y\|_2}$ up to relative error.

We overcome these issues in our analysis by noting that we do not require a relative error approximation! Even non-private KDE data structures (such as those in Table 2) already incur additive errors. This motivates proving additive error approximation results, where the additive error from dimensionality reduction is comparable to the additive error incurred by existing non-private KDE data structures. We accomplish this via a careful analysis of the non-linear kernel functions and show that it is possible to project onto a *constant* dimension, depending on the additive error, which is independent of the original dataset size $n$ or original dimensionality $d$.

**Theorem 2.1** (Informal; see Theorems E.2 and F.2). *Consider the Gaussian and exponential kernels. For any $\alpha \in (0, 1)$, projecting the dataset and query to $\tilde{O}(1/\alpha^2)$ dimensions using an oblivious JL map preserves the KDE value up to additive error $\alpha$. For the Cauchy kernel, projecting to $O(1/\alpha^2)$ dimensions preserves the KDE value up to multiplicative $1 + \alpha$ factor.*

We note that variants of 'dimensionality reduction' have been studied for Gaussian kernels, most notably via *coresets* which reduce the dataset size (one can then correspondingly reduce the dimension by projecting onto the data span; see Luo et al. (2023); Phillips & Tai (2020)). However, these coresets are data-dependent and it is not clear if they respect DP guarantees. On the other hand, our results use random matrices that do not leak privacy. Lastly, our analysis also sheds additional light on the power of randomized dimensionality reduction *beyond* JL for structured problems, complementing a long line of recent works (Boutsidis et al., 2010; Cohen et al., 2015; Becchetti et al., 2019; Makarychev et al., 2019; Narayanan et al., 2021; Izzo et al., 2021; Charikar & Waingarten, 2022).

**Smooth Kernels via Function Approximation Theory.** For DP-KDE, we also exploit low-dimensional structures via function approximation, by approximating kernels such as $\frac{1}{1 + \|x - y\|_2}$ in terms of exponential functions. To be more precise, for $h(x, y) = \|x - y\|_2, \|x - y\|_2^2$, and $\|x - y\|_1$, Corollary 3.2 allows us to express $\frac{1}{1 + h(x, y)} \approx \sum_{j \in J} w_j e^{-t_j h(x, y)}$ for explicit parameters $t_j, w_j$. The corollary follows from a modification of results in approximation theory, see Section 3. This can be viewed as projecting the kernel onto a low-dimensional span of exponential functions, since only $|J| = \tilde{O}(1)$ terms in the sum are required. We can then benefit from already existing KDE data structures for various kernels involving exponential functions, such as the exponential kernel! Hence, we obtain new private KDE queries for a host of new functions in a black-box manner. The fact that $|J|$ (the number of terms in the sum) is small is crucial, as instantiating a differentially

private KDE data structure for each $j \in [J]$ does not substantially degrade the privacy guarantees or construction and query times. This reduction is detailed in Section 3.

**Outline of the Paper.** Due to space constraints, only the function approximation theory reduction is presented in the main body in Section 3. The rest of our theoretical results is deferred to the appendix with full proofs. In Section 4, we empirically verify our upper bound algorithms and give applications to private classification. Our $\ell_1$ algorithm in one dimension is in Section B. It contains the main ideas for the high-dimensional $\ell_1$ algorithm, given in Appendix B.1. Appendix C states our lower bounds for the $\ell_1$ distance function. Applications of our $\ell_1$ upper bound, such as for $\ell_2$ and $\ell_p^p$, are given in Appendix D, and improved DP bounds for the exponential and Gaussian kernels are given in Appendix E. Appendix F contains improved DP results for smooth kernels (such as Cauchy kernels). Appendix G contains our improved KDE query bounds in the non-private setting.

**Related Work.** We use the standard definition of differential privacy (Dwork et al., 2006), given in Definition A.1. We survey the most relevant prior works. Additional related works are given in Section A.2. We write guarantees in terms of the expected error for any fixed query. These algorithms, and ours, can easily be converted to high-probability results by taking the median of multiple (logarithmically many) independent copies. The theorem statement below pertains to the distance functions $\|x - y\|_1$. It is stated for the case where all the dataset points and queries are in the box $[0, 1]^d$, but easily extend to a larger domain by scaling.

**Theorem 2.2** (Huang & Roth (2014)). *Assume the dataset and query points are contained in $[0, 1]^d$. There exists an algorithm which outputs an $\varepsilon$-DP data structure for the function $\|x - y\|_1$ with the following properties: (1) the expected additive error is $\tilde{O}\left(\frac{n^{1/3}d^{7/3}}{\varepsilon^{2/3}}\right)$[1], (2) the construction time is $O\left(n^{8/3}\varepsilon^{2/3}d^{2/3}\right)$, (3) the space usage is $O\left(\frac{n^{2/3}\varepsilon^{2/3}}{d^{1/3}}\right)$, (4) and the query time is $O(d)$.*

(Huang & Roth, 2014) also obtained results for $\ell_2$ with additive errors containing factors of $n$ and $d$ as shown in Table 1; see Appendix A.2 for the formal statement of their results. The result of Wagner et al. (2023) concerns private KDE constructions for the exponential ($e^{-\|x-y\|_2}$), Gaussian ($e^{-\|x-y\|_2^2}$), and Laplace ($e^{-\|x-y\|_1}$) kernels.

**Theorem 2.3** (Wagner et al. (2023)). *Let $\alpha \in (0, 1)$ and suppose $n \geq \Omega\left(\frac{1}{\alpha\varepsilon^2}\right)$. For $h(x, y) = \|x - y\|_2, \|x - y\|_2^2$, or $\|x - y\|_1$, there exists an algorithm which outputs an $\varepsilon$-DP data structure for $f(x, y) = e^{-h(x,y)}$ with the following properties: (1) the expected additive error is at most $\alpha$, (2) the query time is $O\left(\frac{d}{\alpha^2}\right)$, the construction time is $O\left(\frac{nd}{\alpha^2}\right)$, and the space usage is $O\left(\frac{d}{\alpha^2}\right)$.*

Earlier works also study or imply DP KDE. The results of Wagner et al. (2023) were shown to be superior, thus we only explicitly compare to Wagner et al. (2023); see Appendix A.2.

## 3 SPARSE FUNCTION APPROXIMATION

We provide details on the function approximation theory used in Section F to obtain our results on smooth kernels. We use the fact that a small number of exponential sums can approximate smooth kernels, enabling us to reduce this case to prior kernels in Section E. First we recall a classic result.

**Theorem 3.1** (Sachdeva & Vishnoi (2014)). *Given $\varepsilon, \delta \in (0, 1]$, there exist $O(\log(1/(\varepsilon \cdot \delta)))$ positive numbers $w_j, t_j > 0$, all bounded by $O\left(\frac{1}{\varepsilon \log(1/\delta)}\right)$, such that for all $x \in [\varepsilon, 1]$ we have $(1-\delta)x^{-1} \leq \sum_j w_j e^{-t_j x} \leq (1 + \delta)x^{-1}$. Furthermore, $|w_j e^{-t_j}| \leq O(1)$ for all $j$.*

The theorem implies the following useful corollary, proved in Appendix F.2.

**Corollary 3.2.** *Given $\alpha \in (0, 1]$, there exist $O(\log(1/\alpha))$ positive numbers $w_j, t_j > 0$, all bounded by $O\left(\frac{1}{\alpha \log(1/\alpha)}\right)$, such that for all $x \geq 1$ we have $\left|\sum_j w_j e^{-t_j x} - x^{-1}\right| \leq \alpha$.*

---

[1]Throughout the paper, $\tilde{O}$ hides logarithmic factors in $n$ and $d$.

Using Corollary 3.2, we can obtain private KDE data structures for the kernels $f(x,y) = \frac{1}{1+\|x-y\|_2}, \frac{1}{1+\|x-y\|_1}, \frac{1}{1+\|x-y\|_2^2}$ via a black-box reduction to the corresponding private KDE data structures for the kernels $e^{-\|x-y\|_2}, e^{-\|x-y\|_2^2}$, and $e^{-\|x-y\|_1}$.

**Theorem 3.3.** *Let $h(x,y) = \|x-y\|_2, \|x-y\|_2^2$, or $\|x-y\|_1$ and $\alpha \in (0,1)$. Suppose there exists an algorithm for constructing an $\varepsilon$-DP KDE data structure for the kernel $e^{-h(x,y)}$ on a given dataset of size $n$ which answers any query with expected additive error $\alpha$, takes $C(n,\alpha)$ construction time, $Q(n,\alpha)$ query time, and $S(n,\alpha)$ space, assuming $n \geq L(\varepsilon, \alpha)$.*

*Then, there exists an $\varepsilon$-DP data structure for answering KDE queries for $f(x,y) = \frac{1}{1+h(x,y)}$ which answers any query with expected additive error $\alpha$ and the same construction, query, and space as the exponential case, but with $n$ replaced by $O(n\log(1/\alpha))$ and $\alpha$ replaced by $\alpha/\log(1/\alpha)$ in the functions $C, Q, S$, and $L$.*

*Proof.* We give a reduction showing how (a small collection of) private KDE data structures for the kernel $e^{-h(x,y)}$ can be used to answer KDE queries for $f(x,y) = \frac{1}{1+h(x,y)}$. Let $g(z)$ be the function guaranteed by Corollary 3.2 which approximates $1/z$ by an additive factor for all $z \geq 1$:

$$|\underbrace{\sum_j w_j e^{-t_j z}}_{g(z)} - z^{-1}| \leq O(\alpha) \quad \forall z \geq 1. \text{ We have}$$

$$\frac{1}{|X|}\sum_{x\in X} f(x,y) = \frac{1}{|X|}\sum_{x\in X}\frac{1}{1+h(x,y)} = \left(\frac{1}{|X|}\sum_{x\in X}\sum_j w_j e^{-t_j(1+h(x,y))}\right) + O(\alpha)$$

$$= \left[\sum_j w_j e^{-t_j}\left(\frac{1}{|X|}\sum_{x\in X}e^{-t_j h(x,y)}\right)\right] + O(\alpha) = \left[\sum_j w_j e^{-t_j}\left(\frac{1}{|X_j|}\sum_{x\in X_j}e^{-h(x,y_j)}\right)\right] + O(\alpha)$$

where $X_j$ is the dataset $X_j = \{t_j \cdot x \mid x \in X\}$ and $y_j$ is the query $t_j \cdot y$ in the cases that $h(x,y) = \|x-y\|_1$ or $\|x-y\|_2$. In the case where $h(x,y) = \|x-y\|_2^2$ we have $X_j = \{\sqrt{t_j}\cdot x \mid x \in X\}$ and $y_j$ is the query $\sqrt{t_j}\cdot y$.

Note that the function $g$ is public so the parameters $w_j$ and $t_j$ are publicly known (and do not depend on the dataset). Now we simply instantiate private KDE data structures which approximate each of the sums $\frac{1}{|X_j|}\sum_{x\in X_j}e^{-h(x,y)}$. More specifically, we release $O(\log(1/\alpha))$ kernel KDE data structures, one for each $X_j$, and each of which is $O(\varepsilon/\log(1/\alpha))$-DP. Then the overall data structures we release are $\varepsilon$-DP by composition. Furthermore, since each $w_j e^{-t_j} = O(1)$ and there are only $O(\log(1/\alpha))$ of these terms, if each data structure has expected additive error $O(\alpha/\log(1/\alpha))$, then the overall error is $O(\alpha)$ as well. To summarize, the logarithmic blowup happens in the query/space, as well as any lower-bound assumption on the size of the dataset. $\square$

## 4 EMPIRICAL EVALUATION

We evaluate our algorithms on synthetic and real datasets. We consider three experimental settings which together are representative of our main upper-bound results. We show the average of 20 trials and $\pm 1$ standard deviation is shaded where appropriate. All experiments unless stated otherwise are implemented in Python 3.9 on an M1 MacbookPro with 32GB of RAM.

$\ell_1$ **Experiments.** The task here is to approximate the (normalized) map $y \to \frac{1}{n}\sum_{x\in X}|x-y|$ for a one dimensional dataset $X$ of size $n = 10^3$, with points randomly picked in $[0,1]$. The query points are $O(n)$ evenly spaced points in $[0,1]$. We implement our one-dimensional $\ell_1$ distance query algorithm and compare to the prior baseline of (Huang & Roth, 2014). Both our and (Huang & Roth, 2014)'s high-dimensional algorithms are constructed by instantiating $d$ different copies of one-dimensional data structures (on the standard coordinates ). Thus, the performance in one dimension is directly correlated with the high-dimensional case. In our case, the map we wish to approximate converges to $\int_0^1 |x-y|\,dx = y^2 - y + 1/2$ for $y \in [0,1]$, allowing us to easily compare to the ground truth. In Figure 1a, we plot the average relative error across all queries as

a function of $\varepsilon$. The explicit parameter settings for the algorithm given in (Huang & Roth, 2014) are extremely large in practice, meaning the output of their algorithm was always the identically 0 function, which gives relative error equal to 1 (the distance query was always estimated to be 0) for the values of $\varepsilon$ tested, as shown in Figure 1a. On the other hand, our algorithm gives non-trivial empirical performance and its error decreases smoothly as $\varepsilon$ increases. Indeed, Figure 1b shows our output values (scaled by $1/n$) for various $\varepsilon$'s. We can observe that our estimates converge to the true function as $\varepsilon$ increases. We observed qualitatively similar results for for $n = 10^6$ as well.

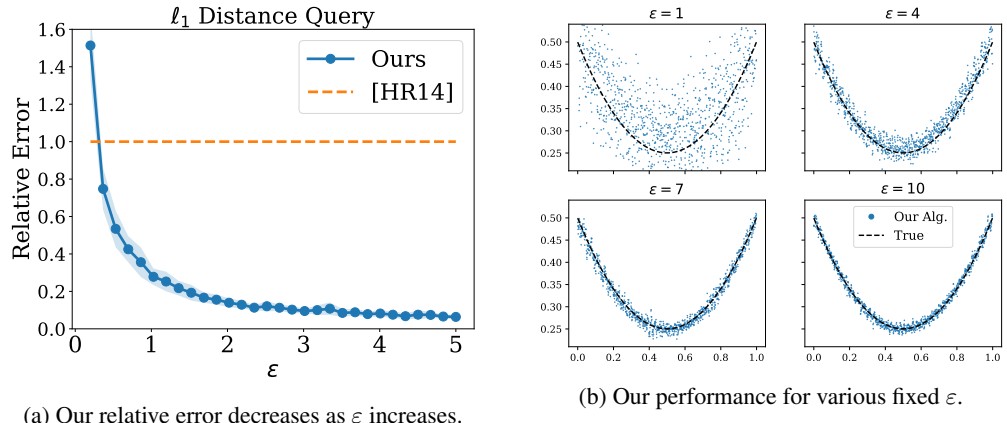

(a) Our relative error decreases as $\varepsilon$ increases.

(b) Our performance for various fixed $\varepsilon$.

Figure 1: Our algorithm for $\ell_1$ queries has smaller error than prior SOTA of (Huang & Roth, 2014).

**Dimensionality Reduction Experiments.**   We empirically demonstrate that dimensionality reduction provides computational savings for DP-KDE without significantly degrading accuracy. Our task is to approximate KDE values for the Gaussian kernel $e^{-\|x-y\|_2^2}$. We compare against the prior SOTA (Wagner et al., 2023). Our provable dimensionality reduction result of Theorem E.2 gives a general framework: apply an oblivious dimensionality reduction to the data and use any DP-KDE algorithm in the projected space. Indeed, Theorem E.1 follows by applying the framework to the prior SOTA algorithm of Wagner et al. (2023). Thus in our experiment, we use the randomized dimension reduction of Theorem E.2 in conjunction with the implementation of Wagner et al. (2023). Note that while we fix the DP-KDE implementation used after dimensionality reduction, our framework is compatible with any other choice and we expect qualitatively similar results with other choices.

Our dataset consists of embeddings of CIFAR-10 in dimension 2048, computed from an Inception-v3 model (Szegedy et al., 2016), pre-trained on ImageNet (Deng et al., 2009). Obtaining embeddings of private datasets from pre-trained ML models is standard in the applied DP literature (Yu et al., 2020; De et al., 2022). The intuition is that the embeddings from the network are powerful enough to faithfully represent the images in Euclidean space, so computing kernel values on these features is meaningful. We project the embeddings to lower dimensions $d$ ranging from 200 to 2000. We use the training points of a fixed label as the private dataset and the corresponding test set as the queries.

Figure 2a shows the relative error of our approach (in blue) and the baseline of Wagner et al. (2023) (in orange) which does not use any dimensionality reduction. The relative errors of both are computed by comparing to the ground truth. Figure 2b shows the construction time of the private data structure and Figure 2c shows the total query time on the test points. We see that the relative error smoothly decreases as we project to more dimensions, while construction and query time smoothly increase. Note the construction time includes the time to compute the projection. For example, projecting to $d = 1000$ increases the relative error by 0.015 in absolute terms, while reducing the construction time by $\approx$ 2x and reducing the construction time by a factor of > 4x.

**Differentially Private Classification.**   We consider the DP classification task on Cifar-10. The train and test splits are the private data and query respectively, and the task is to train a private classifier on the train set to classify the test set. Our methodology is extremely simple, fast, and does not require any specialized hardware like GPUs: we instantiate an $(\varepsilon, \delta)$-DP distance query data structure on each class. The classes disjointly partition the data so the output, which is an $(\varepsilon, \delta)$-DP data structure for each class, is overall $(\varepsilon, \delta)$-DP (Ponomareva et al., 2023). We use

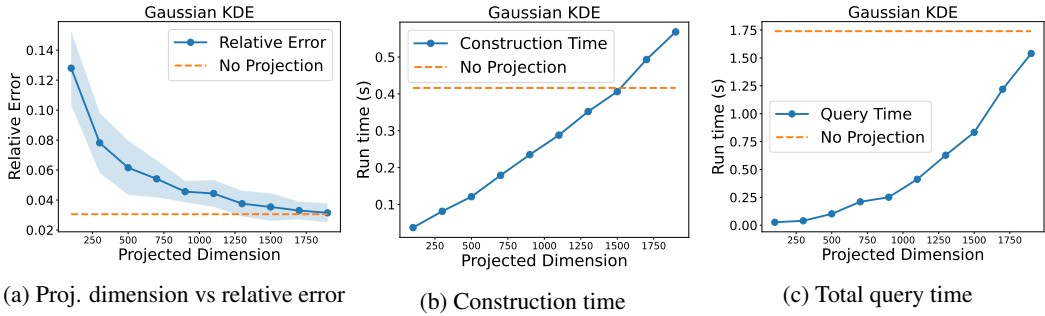

(a) Proj. dimension vs relative error      (b) Construction time      (c) Total query time

Figure 2: Results for our dimensionality reduction experiments.

$f(x, y) = \|x - y\|_2^2$ since it has the simplest algorithm (see Section D.1). It essentially reduces to releasing a noisy mean for every class. To classify a query, we assign it to the class whose (noisy) mean is closest to the query. For other potential choices of $f$ like kernels, one would instead assign to the class with the highest KDE value. There are two competing SOTA works: one from Deepmind (De et al., 2022) and (Yu et al., 2020). We give a high-level methodology of prior works (see Appendix H for more details): they start with a pre-trained model on ImageNet[2] and fine tune using DP-gradient descent/SGD. Their specific model details are given in Appendix H. Note that the vectors we build our private data structures on are the penultimate layer embeddings of the ResNet pre-trained model used in Yu et al. (2020). Thus, all methods have the access to the same 'pre-training' information.

Let us temporarily ignore $\delta$ for simplicity of discussion. If they use $T$ steps of training, every model update step approximately satisfies $\varepsilon/\sqrt{T}$ privacy (exact bounds depend on the DP composition formulas), ensuring the overall final model is $(\varepsilon, \delta)$-DP. Thus, every step for them incurs some privacy budget, with the benefit of increased accuracy. Therefore, these works can stop training at intermediate times to obtain a model with stronger privacy guarantees (lower $\varepsilon$), but worse accuracy. However, the same procedure also gives an accuracy vs model training time trade-off for these prior works. This is shown in Figure 3. The right most endpoints of both baselines (the largest times), correspond to models with $\varepsilon = 1$. In other words, their models with the worst privacy guarantees has the highest accuracy, while also requiring the longest training time.

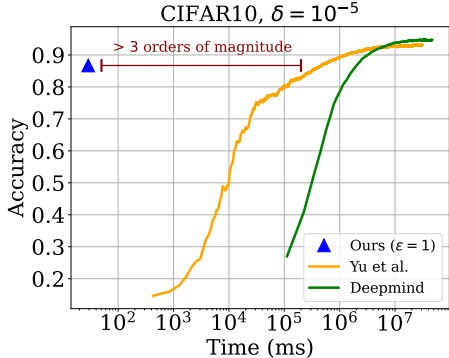

Figure 3: Runtime vs Accuracy.

In contrast, our algorithm of Section D.1 simply releases a noisy collection of fixed vectors. Our data structure construction *time*, which corresponds to their model training time, is independent of $(\varepsilon, \delta)$ (but *accuracy* depends on them). In Figure 3, we plot the accuracy of our data structure for $\varepsilon = 1$ (we use the best hyper-parameter choices for all methods). For other values of $\varepsilon$, we would simply incur the same construction time, but observe differing accuracy since the construction time is independent of $\varepsilon$ (but again accuracy improves for higher $\varepsilon$). The time to initialize our data structure (for all classes) is 28.8 ms *on a CPU*, and the query time for all queries was 73.8 ms. On the other hand, fully training the model of Yu et al. (2020) up to $\varepsilon = 1$ takes $> 8.5$ hours on a single NVIDIA RTX A6000 GPU. The runtimes of De et al. (2022) are even longer since they use larger models. Thus, as shown in Figure 3, creating our private data structure is $> $ **3 orders of magnitude** faster than the time to create models of corresponding accuracy via the two baselines. Note that we are also using arguably inferior hardware. The best accuracy of all methods as a function of $\varepsilon$, ignoring run-time considerations, is shown in Figure 4.

---

[2]The works consider other alternate datasets, but we only compare to the Imagenet case. We expect qualitatively similar results when pre-training with other datasets.

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

## A   ADDITIONAL PRELIMINARIES

### A.1   STANDARD DIFFERENTIAL PRIVACY RESULTS

Two datasets $X, X'$ are neighboring if they differ on a single data point.

**Definition A.1** (Dwork et al. (2006)). *Let M be a randomized algorithm that maps an input dataset to a range of outputs $\mathcal{O}$. For $\varepsilon, \delta > 0$, M is defined to be $(\varepsilon, \delta)$-DP if for every neighboring datasets $X, X'$ and every $O \subseteq \mathcal{O}$, $\Pr[M(X) \in O] \leq e^{\varepsilon} \cdot \Pr[M(X') \in O] + \delta$. If $\delta = 0$, we say that M is $\varepsilon$-DP (this is called pure differential privacy).*

**Theorem A.1** (Advanced Composition Starting from Pure DP (Dwork et al., 2010)). *Let $M_1, \ldots, M_k : \mathcal{X}^n \to \mathcal{Y}$ be randomized algorithms, each of which is $\varepsilon$-DP. Define $M : \mathcal{X}^n \to \mathcal{Y}^k$ by $M(x) = (M_1(x), \ldots, M_k(x))$ where each algorithm is run independently. Then M is $(\varepsilon', \delta)$-DP for any $\varepsilon, \delta > 0$ and*

$$\varepsilon' = \frac{k\varepsilon^2}{2} + \varepsilon\sqrt{2k \log(1/\delta)}.$$

*For $\delta = 0$, M is $k\varepsilon$-DP.*

### A.2   ADDITIONAL RELATED WORKS

**Distance Query Related Works.**   The construction times and query times in the result below are not explicitly stated in Huang & Roth (2014), but they are likely to be similar to that of Theorem 2.2.

**Theorem A.2** (Huang & Roth (2014)). *Assume the dataset and query points are contained in the $\ell_2$ ball of diameter 1. There exists an algorithm which outputs an $\varepsilon$-DP data structure for the distance function $f(x, y) = \|x - y\|_2$ such that the expected additive error is $\tilde{O}\left(\frac{n^{15/17} d^{7/17}}{\varepsilon^{2/17}}\right)$.*

Feldman et al. (2009) give lower bounds for additive errors of private algorithms which approximate the $k$-median cost function, which is related to the $\ell_1$ distance query. However, their lower bound only applies to coresets specifically, whereas our lower bounds hold for any private mechanism. There have also been recent works designing scalable algorithms for computing distance functions in the non-private setting; see Indyk & Silwal (2022) and references therein.

**KDE Related Works.**   Earlier works such as the mechanisms in (Gupta et al., 2012; Blum et al., 2013; Aldà & Rubinstein, 2017) also study or imply results for DP-KDE. However, many suffer from drawbacks such as exponential dependency on $d$ for running time. The results of Wagner et al. (2023) were shown to be superior to such prior methods (see therein for more discussions), so we only compare to the current state of the art DP KDE results from Wagner et al. (2023).

**Generic Private Queries.**   We refer the reader the the excellent survey of Vadhan (2017a) and references therein for an overview of algorithms and discussions for broad classes of queries for private datasets. Lastly, we note that dimensionality reduction has been studied in differential privacy in non-KDE contexts in (Blocki et al., 2012; Singhal & Steinke, 2021); see the references therein for further related works.

## B   $\ell_1$ DISTANCE QUERY

We construct a private data structure for answering $\ell_1$ distance queries in one dimension. The general high-dimensional case, given fully in Appendix B.1, can be handled as follows: create a collection of $d$ one-dimensional data structures, constructed on the standard coordinate projections of the dataset. We now describe our one-dimensional algorithm. For the sake of simplicity, let us assume for now that all dataset points are integer multiples of $1/n$ in $[0, 1]$. This holds without loss of generality as shown later. Furthermore, let us also instead consider the slightly different *interval* query problem. Here we are given an interval $I \subset \mathbb{R}$ as the query, rather than a point $y$, and our data structure outputs $|I \cap X|$. We can approximate a distance query at any $y$ by asking appropriate interval queries $I$, for example geometrically increasing intervals around the query point $y$.

To motivate the algorithm design, let us additionally ignore privacy constraints for a moment. We use the classic binary tree in one dimension: its leaves correspond to the integer multiples of $1/n$ in $[0, 1]$ and store the number of dataset points in that particular position, while internal nodes store the sum of their children. It is well-known that any interval query $I$ can be answered by adding up the values of only $O(\log n)$ tree nodes. To handle privacy, we release a noisy version of the tree. We note that changing any data point can only change $O(\log n)$ counts in the tree, each by at most one (the leaf to root path). This bounds the sensitivity of the data structure. The formal algorithm and guarantees are stated below. Before presenting them, we make some simplifications which hold without loss of generality.

**Remark B.1** (Simplifications). *(1) We scale all dataset points from $[0, R]$ to $[0, 1]$ by dividing by $R$. We also scale $y$. We can undo this by multiplying our final estimate by $R$. (2) After scaling, we assume $y \in [0, 1]$. If $y$ is outside $[0, 1]$, for example if $y \geq 1$, we can just instead query $1$ and add $n(y - 1)$ to the final answer, since all dataset points are in $[0, 1]$. This does not affect the approximation. (3) Lastly, we round all points to integer multiples of $1/n$, introducing only $O(R)$ additive error.*

---

**Algorithm 1** Pre-processing data structure

1: **Input:** A set $X$ of $n$ numbers in the interval $[0, 1]$, privacy parameter $\varepsilon$
2: **Output:** An $\varepsilon$-DP data structure
3: **procedure** PREPROCESS
4:      Round every dataset point to an integer multiple of $1/n$
5:      Compute the counts of the number of dataset points rounded to every multiple of $1/n$
6:      Build a balanced binary tree $T$ where internal nodes store the sum of the counts of their children and leaf nodes store the counts of the multiples of $1/n$
7:      Independently add noise drawn from Laplace($\eta$) where $\eta = O(\log(n)/\varepsilon)$ to every count
8:      **Return** tree $T$
9: **end procedure**

---

**Algorithm 2** Interval Query

1: **Input:** Tree $T$, interval $Q \subseteq [0, 1]$
2: **procedure** NOISYCOUNT
3:      Round the endpoints of $Q$ to the closest multiple of $1/n$
4:      Break $Q$ up into the smallest number of contiguous and disjoint pieces such that there is a node in $T$ representing each piece          ▷ *At most $O(\log n)$ pieces are required*
5:      **Return** the sum of the counts in each of the nodes in $T$ computed above
6: **end procedure**

---

**Algorithm 3** One dimensional Distance Query

1: **Input:** data structure $T$ from Algorithm 1, query $y \in [0, 1]$, accuracy parameter $\alpha \in (0, 1)$.
2: **procedure** DISTANCEQUERY
3:      Round $y$ to the closest integer multiple of $1/n$
4:      Value $\leftarrow 0$
5:      **for** $j = 0, 1, ..., O(\log(n)/\alpha)$ **do**
6:          $Q_j \leftarrow \left[ y + \frac{1}{(1+\alpha)^{j+1}}, y + \frac{1}{(1+\alpha)^j} \right)$      ▷ *This will consider the points to the right of $y$*
7:          Value $\leftarrow$ Value $+$ NoisyCount($Q_j$) $\cdot \frac{1}{(1+\alpha)^j}$
8:      **end for**
9:      Repeat the previous loop for intervals to the left of $y$
10:      **Return** Value
11: **end procedure**

---

**Lemma B.1.** *The tree $T$ returned by Algorithm 1 is $\varepsilon$-DP.*

*Proof.* We can encode the tree $T$ as a vector in dimension $O(n)$. Changing one input data point only changes $O(\log n)$ entries of this vector, each by 1, thus the sensitivity of $T$ is $O(\log n)$. Adding

coordinate-wise Laplace noise of magnitude $\eta = O(\log(n)/\varepsilon)$ suffices to ensure $\varepsilon$-DP using the standard Laplace mechanism. □

We now analyze the utility of the algorithm.

**Theorem B.2.** *Suppose $X \subseteq [0, R]$ is a dataset of $n$ numbers in one dimension. Let $\alpha \in (0, 1)$ be the accuracy parameter used in Algorithm 3. Let $A$ be the output of Algorithm 3 and let $A' = \sum_{x \in X} |y - x|$ be the true distance query value. Then we have $\mathbb{E}\,|A - A'| \leq \alpha A' + \tilde{O}\left(\frac{R}{\varepsilon\sqrt{\alpha}}\right)$.*

*Proof.* For simplicity, we only consider the distance query to the points in $X$ to the right of $y$. The identical proof extends to the symmetric left case. We also work under the simplifications stated in Remark B.1. They only affect the additive error by at most $O(R)$. For an interval $Q$, define TrueCount($Q$) to be the true value $|Q \cap X|$. Let

$$\text{Estimate}_1 = \sum_{j \geq 0} \frac{1}{(1+\alpha)^j} \cdot \text{TrueCount}(Q_j) \qquad \text{and} \qquad A' = \sum_{j \geq 0} \sum_{x \in X \cap Q_j} |y - x|.$$

First, we know that $|\text{Estimate}_1 - A'| \leq \alpha \cdot A'$, as for all $j$, the distanced between $y$ and different points $x \in X \cap Q_j$ only differ by a multiplicative $(1 + \alpha)$ factor. Thus, it suffices to show the output as returned by Algorithm 3, i.e. $A$, differs from Estimate$_1$ by $\tilde{O}\left(\frac{1}{\varepsilon\sqrt{\alpha}}\right)$. Let NoisyCount($Q$) denote the interval query answer returned by our noised-tree via Algorithm 2. Algorithm 3 outputs $A = \sum_{j \geq 0} \frac{1}{(1+\alpha)^j} \cdot \text{NoisyCount}(Q_j)$. We wish to bound

$$|\text{Estimate}_1 - A| \leq \left| \sum_{j \geq 0} \frac{1}{(1+\alpha)^j} \cdot (\text{TrueCount}(Q_j) - \text{NoisyCount}(Q_j)) \right|.$$

Note that $Z_j := \text{TrueCount}(Q_j) - \text{NoisyCount}(Q_j)$ is equal to the sum of at most $O(\log n)$ Laplace random variables, each with parameter $O((\log n)/\varepsilon)$. This is because we compute all noisy counts by accumulating the counts stored in the individual nodes in $T$ corresponding to $Q_j$. We only query $O(\log n)$ nodes for any $Q_j$ and each node has independent noise added. Thus, $\mathbb{E}\,Z_j = 0$ and $\text{Var}\left[Z_j \cdot \frac{1}{(1+\alpha)^j}\right] \leq \frac{\tilde{O}(1)}{\varepsilon^2} \cdot \frac{1}{(1+\alpha)^{2j}}$. In addition, the $Z_j$'s are also independent of each other since the intervals $Q_j$'s are disjoint, meaning we query disjoint sets of nodes in the tree for different $Q_j$'s. Hence,

$$\text{Var}\left[\sum_{j \geq 0} \frac{1}{(1+\alpha)^j} \cdot Z_j\right] \leq \frac{\tilde{O}(1)}{\varepsilon^2} \cdot \sum_{j \geq 0} \frac{1}{(1+\alpha)^{2j}} \leq \frac{\tilde{O}(1)}{\alpha\varepsilon^2}, \tag{1}$$

meaning with large constant probability, say at least 0.999, the quantity $|\text{Estimate}_1 - A|$ is at most $\tilde{O}(1)/(\varepsilon\sqrt{\alpha})$ by Chebyshev's inequality. A similar conclusion also holds in expectation since for any centered random variable $W$, $\mathbb{E}\,|W| \leq \sqrt{\text{Var}(W)}$. We recover our desired statement by multiplying through by $R$ to undo the scaling. □

## B.1 HIGH-DIMENSIONAL $\ell_1$ QUERY

Algorithm 3 automatically extends to the high dimensional case due to the decomposability of the $\ell_1$ distance function. Indeed, we simply instantiate $d$ different one-dimensional distance query data structures, each on the coordinate projection of our private dataset. The algorithm is stated below. For simplicity, we state both the preprocessing and query algorithms together.

---

**Algorithm 4** High-dimensional $\ell_1$ distance query

---

1: **Input:** Set $X$ of $n$ $d$-dimensional points in the box $[0, R]^d$, privacy parameter $\varepsilon$, mulitplicative accuracy parameter $\alpha$, query $y$
2: **procedure** $\ell_1^d$ QUERY
3:     Instantiate $d$ different one-d data structures $\mathcal{D}_1, \ldots, \mathcal{D}_d$. $\mathcal{D}_i$ is the output of Algorithm 1 on the $i$th coordinate projections of $X$. Each data structure is $\varepsilon/d$-DP      ▷ *Preprocessing Stage*
4:     **Return** The sum of outputs when $\mathcal{D}_i$ is queried on $y_i$ for every $i$      ▷ *Query Stage*
5: **end procedure**

---

The following theorem is a corollary of Lemma B.1 and Theorem B.2.

**Theorem B.3.** *Let $A$ be the output of Algorithm 4. Let $A' = \sum_{x \in X} \|y - x\|_1$ be the true answer. We have $\mathbb{E}\,|A - A'| \leq \alpha A' + \tilde{O}\left(\frac{Rd^{1.5}}{\varepsilon\sqrt{\alpha}}\right)$. Furthermore, Algorithm 4 is $\varepsilon$-DP.*

*Proof.* The $\varepsilon$-DP guarantee follows from standard DP composition results (Theorem A.1), so it remains to argue about the approximation guarantee. Let $A_i$ be the estimate returned by $\mathcal{D}_i$ and let $A_i'$ be the true answer in the $i$th dimension. Note that $A' = \sum_i A_i'$ and $A = \sum_i A_i$. Naively applying Theorem B.2 gives us additive error $\tilde{O}(Rd^2/(\varepsilon\sqrt{\alpha}))$. However, we can exploit the fact that the data structures in the individual dimensions are using independent randomness to get a better bound.

Let us inspect the proof of Theorem B.2. Let $Z_j^i$ be the variables $Z_j$ used in the proof of Theorem B.2 for coordinate $i$. Similar to the proof of Theorem B.2, we can note that the error incurred by our estimate among *all* coordinates, can be upper bounded by the absolute value of $\sum_j \frac{1}{(1+\alpha)^j}\left(\sum_i Z_j^i\right)$, where each $Z_j^i$ are independent across $i$ and $j$ and are each the sum of at most $O(\log n)$ different Laplace $O((\log n)/\varepsilon)$ random variables. The variance of each individual dimension is given by Equation 1 (with $\varepsilon$ scaled down by $\varepsilon/d$), i.e., it is of the order $\tilde{O}(d^2)/(\alpha\varepsilon^2)$. The total variance across $d$ copies is then $\tilde{O}(d^3)/(\alpha\varepsilon^2)$. Finally, the same calculations as the proof of Theorem B.2 imply an additive error of the square root of this quantity, namely $\tilde{O}(d^{1.5})/(\sqrt{\alpha}\varepsilon)$. $\qquad\square$

If we relax the privacy guarantees to approximate DP, we can get a better additive error, matching the additive error term in the lower bound of Theorem C.2 (note however that Theorem C.2 is a lower bound on pure DP algorithms, not approximate DP).

**Theorem B.4.** *Let $\delta > 0$ and $A$ be the output of Algorithm 4 where every one dimensional algorithm is instantiated to be $(c\varepsilon/\sqrt{d\log(1/\delta)})$-DP for a sufficiently small constant $c$ independent of all parameters. Let $A' = \sum_{x \in X} \|y - x\|_1$ be the true answer. We have $\mathbb{E}\,|A - A'| \leq \alpha A' + \tilde{O}\left(\frac{Rd\sqrt{\log(1/\delta)}}{\varepsilon\sqrt{\alpha}}\right)$. Furthermore, Algorithm 4 is $(\varepsilon, \delta)$-DP assuming $\varepsilon \leq O(\log(1/\delta))$. $\tilde{O}$ hides logarithmic factors in $n$.*

*Proof.* The same proof as Theorem B.3 applies, but instead we use the approximate DP advanced composition result A.1, and an appropriately smaller noise parameter in Algorithm 1. $\qquad\square$

## C  LOWER BOUNDS FOR $\ell_1$ DISTANCE QUERIES

First we obtain lower bounds for the one-dimensional case, which is then extended to the arbitrary dimensional case. Recall the definition of the dimensional distance query problem: Given a dataset $X$ of $n$ points in the interval $[0, R]$, an algorithm outputs a data structure which given query $y \in \mathbb{R}$, computes the value $\sum_{x \in X} |x - y|$. Our lower bound idea is via the 'packing lower bound' technique used in DP (Hardt & Talwar, 2010; Beimel et al., 2014; Vadhan, 2017b). At a high level, we construct many datasets which differ on very few points. By the restrictions of DP, the $\ell_1$ distance queries on these datasets must be similar, since they are all 'nearby' datasets. However, our construction will ensure that these different datasets result in vastly different *true* $\ell_1$ distance queries for a fixed set of queries. This implies a lower bound on the additive error incurred from the privacy requirements.

**Theorem C.1.** *For sufficiently large $n$ and any $\varepsilon < 0.2$, any $\varepsilon$-DP algorithm which outputs a data structure such that with probability at least $2/3$, the distance query problem is correctly answered on any query $y$ with additive error at most $T$, must satisfy $T = \Omega(R/\varepsilon)$.*

*Proof.* Let $T$ be the additive error of an algorithm $\mathcal{A}$ as described in the theorem statement. Our goal is to show that we must have $T \geq \Omega(R/\varepsilon)$. Note that crucially we know the value of $T$. Since the purported algorithm outputs an $\varepsilon$-DP data structure, we use the value of $T$ to design a 'hard' instance for the algorithm.

Let $\alpha \in [0, 1]$ be a parameter. Define $2^{R^{1-\alpha}}$ different datasets as follows: we first put markers in the interval $[0, R]$ at locations $kR^\alpha$ for all $0 \leq k \leq R^{1-\alpha}$. At every marker besides 0, we either put 0

or $\gamma := \lceil \frac{3T}{R^\alpha} \rceil$ data points, and consider all such possible choices. The rest of the data points are put at location $0$. For sufficiently large $n$, this results in the claimed number of datasets.

Now for every such dataset $D$, define the function $f_D : \mathbb{R} \to \mathbb{R}$ defined as

$$f_D(y) = \sum_{x \in D} |x - y|.$$

We claim that the (exact) vector of evaluations

$$[f_D(0), f_D(R^\alpha), f_D(2R^\alpha), \ldots, f_D(R), f_D(R + R^\alpha)]$$

uniquely determines $D$. Indeed, $f_D$ is a piece wise linear function consisting of at most $R^{1-\alpha} + 2$ pieces. Its slopes can only change precisely at the locations $kR^\alpha$. Thus, exactly calculating $f_D((k+1)R^\alpha) - f_D(kR^\alpha)$ gives us the exact values of the slopes, and thus allows us to reconstruct the piece wise linear functions that comprise $f_D$. Correspondingly, this allows us to determine which markers contain a non zero (i.e. $\gamma$) number of points, reconstructing $D$.

The second claim is that the vector of evaluations with entry wise additive error at most $T$ allows for the *exact* reconstruction of the vector of evaluations. This follows from the fact that the exact evaluation values are multiples of $\gamma R^\alpha$ and the additive error is small enough to determine the correct multiple. Formally, we have that $T < \frac{1}{2}\gamma R^\alpha$ and since each $f_D(kR^\alpha)$ is a multiple of $\gamma R^\alpha$, any entry of a noisy evaluation vector with additive error at most $T$ can be easily rounded to the correct value, as it lies closest to a *unique* multiple of $\gamma R^\alpha$.

Now the rest of the proof proceeds via the 'packing' argument for proving lower bounds in differential privacy. Let $Q$ be the queries defined above and let $P_D$ be the set of allowable vectors of evaluations (i.e. those that achieve entry wise error of at most $T$) for dataset $D$ on $Q$. As argued above, the probability that $\mathcal{A}$ on dataset $D$ outputs a vector in $P_D$ is at least $2/3$, and all these sets $P_D$ are disjoint as argued above. Furthermore, all datasets differ in at most

$$\gamma R^{1-\alpha} \leq 3T R^{1-2\alpha} + R^{1-\alpha}$$

data points. Let $D'$ be the dataset with all points at $0$. Group privacy gives us

$$
\begin{aligned}
1 &\geq \sum_D \Pr(\mathcal{A}(D', Q) \in P_D) \\
&\geq \sum_D e^{-(3TR^{1-2\alpha} + R^{1-\alpha})\varepsilon} \cdot \Pr(\mathcal{A}(D, Q) \in P_D) \\
&\geq \sum_D \frac{2}{3} e^{-(3TR^{1-2\alpha} + R^{1-\alpha})\varepsilon} \\
&\geq 2^{R^{1-\alpha}} \cdot \frac{2}{3} e^{-(3TR^{1-2\alpha} + R^{1-\alpha})\varepsilon}.
\end{aligned}
$$

It follows that

$$3T R^{1-2\alpha} + R^{1-\alpha} \geq \frac{\log(2) R^{1-\alpha}}{\varepsilon} + \frac{\log(2/3)}{\varepsilon} \implies T \geq \frac{\log(2) R^\alpha}{3\varepsilon} + \frac{\log(2/3)}{3\varepsilon R^{1-2\alpha}} - \frac{R^\alpha}{3}.$$

Taking $\alpha \to 1$, we can check that for $\varepsilon \leq 0.2$, $T \geq 0.02 R/\varepsilon = \Omega(R/\varepsilon)$, as desired. $\qquad \square$

Let us now extend the lower bound to $d$ dimensions. Recall the problem we are interested in is the following: Given a dataset $X$ of $n$ points in $\mathbb{R}^d$ with every coordinate in the interval $[0, R]$, give an algorithm which outputs a data structure, which given a query $y \in \mathbb{R}^d$, computes the value $\sum_{x \in X} \|x - y\|_1$. The goal is to prove that $\approx Rd/\varepsilon$ additive error is required for answering queries of this form. For a vector $v$, let $v(j)$ denote its $j$th coordinate.

**Theorem C.2.** *For sufficiently large $n$ and $R$ as a function of $d$ and sufficiently small constant $\varepsilon$, any $\varepsilon$-DP algorithm which outputs a data structure which with probability at least $2/3$ answers the above query problem for any query with additive error at most $T$, must satisfy $T = \tilde{\Omega}(Rd/\varepsilon)$.*

*Proof.* We reduce the 1 dimensional version of the problem to the $d$ dimensional version which allows us to use the lower bound of Theorem C.1. Pick $\alpha$ such that $(Rd)^\alpha = Rd/\log(Rd)$ and

suppose $R$ satisfies $R \geq 2^{Cd}$ for a sufficiently large constant $C$. Furthermore, assume that $R$ is an integer multiple of $R^{\alpha}$. Now consider the lower bound construction from Theorem C.1 where the parameter '$R$' there is replaced by $Rd$ and $\alpha$ is as stated. Theorem C.1 implies that an $\varepsilon$-DP data structure which with probability at least $2/3$ correctly answers any distance query for a *one dimensional* input $y$ must have additive error at least $\Omega((Rd)^{\alpha}/\varepsilon) = \tilde{\Omega}(Rd/\varepsilon)$. We will now show how to simulate any one dimensional query $y$ on this lower bound instance with *one $d$* dimensional query on a related $d$ dimensional instance.

To construct the $d$ dimensional instance, consider the interval $[0, Rd]$ as $d$ different blocks, separated by integer multiples of $R$ as $[0, R), [R, 2R), \ldots$ etc. Note that in the one dimensional hard instance we are considering from Theorem C.1, we can always ensure that every one of these $d$ blocks contains the same number of points (For example by only considering such 'balanced' allocations of dataset constructions in the marker construction from C.1. Due to our choice of $R$ and $\alpha$, it is easy to see that the number of such balanced allocations is at least $2^{\Theta((Rd)^{1-\alpha})}$). Let $X_1$ be this one dimensional dataset and let $n'$ be the number of (one dimensional) points that are contained within each block. Consider the $d$ dimensional dataset on $n'$ points where the (one dimensional) points in the first block are the first coordinate projections of the dataset and in general, the points in the $i$th block are the $i$th coordinate projections of the dataset. Since every block has the same number of points, we can construct such a dataset which is consistent with respect to these coordinate projections[3]. Denote this dataset by $X_d$. For a one dimensional query $y$, make a vector $\hat{y} \in \mathbb{R}^d$ which just has $y$ copied in all its coordinates.

We have

$$\sum_{x \in X_1} |y - x| = \sum_{\text{blocks } b} \sum_{x \in b} |y - x|$$

$$= \sum_{j=1}^{d} \sum_{x \in X_d} |x(j) - \hat{y}(j)|$$

$$= \sum_{x \in X_d} \|x - \hat{y}\|_1.$$

Thus, the exact value of the single $d$ dimensional query we have constructed is equal to the exact value of the one dimensional query we are interested in. This reduction immediately implies that any $\varepsilon$-DP data structure which with probability at least $2/3$ answers all $d$ dimensional queries with additive error at most $T$ must satisfy $T = \tilde{\Omega}(Rd/\varepsilon)$, as desired. $\qquad\square$

**Remark C.1.** *The lower bound is slightly stronger than stated since we only assume the query vectors have their one dimensional coordinates bounded by $\tilde{O}(R)$.*

**Remark C.2.** *There is a gap between our upper and lower bounds. Our $\varepsilon$-DP data structure has a $O(d^{1.5})$ dependency whereas the lower bound we prove only states $\Omega(d)$ dependency is required. Note that our approx-DP result of Theorem B.4 has only $O(d)$ dependency, but the lower bound we proved only applies to pure DP algorithms. It is an interesting question to close this gap between the upper and lower bounds.*

## D    COROLLARIES OF OUR $\ell_1$ DATA STRUCTURE

Our high dimensional $\ell_1$ distance query result implies a multitude of downstream results for other distances and functions. For example, it automatically implies a similar result for the $\ell_2$ case via a standard oblivious mapping from $\ell_2$ to $\ell_1$. A similar procedure was applied in Huang & Roth (2014), but using our version of the $\ell_1$ distance query obtains superior results for the $\ell_2$ case. The guarantees of the mapping is stated below.

**Theorem D.1** (Matousek (2002)). *Let $\gamma \in (0, 1)$ and define $T : \mathbb{R}^d \to \mathbb{R}^k$ by*

$$T(x)_i = \frac{1}{\beta k} \sum_{j=1}^{d} Z_{ij} x_j, \quad i = 1, \ldots, k$$

---

[3]Note there are many ways to assign the coordinate projections to the points in $\mathbb{R}^d$. We can just consider one fixed assignment.

*where $\beta = \sqrt{2/\pi}$ and $Z_{ij}$ are standard Gaussians. Then for every vector $x \in \mathbb{R}^d$, we have*

$$\Pr[(1-\gamma)\|x\|_2 \le \|T(x)\|_1 \le (1+\gamma)\|x\|_2] \ge 1 - e^{-c\gamma^2 k},$$

*where $c > 0$ is a constant.*

The mapping $T$ given by Theorem D.1 is *oblivious* to the private dataset and can be released for free without any loss in privacy, and the distances are preserved up to a $1 + \alpha$ multiplicative error if we take $k = O(\log(n)\log(1/\alpha)/\alpha^2)$ for a dataset of size $n$.

**Corollary D.2.** *Let $X$ be a private dataset of size $n$ with a bounded diameter of $R$ in $\ell_2$. There exists an $\varepsilon$-DP data structure such that for any fixed query $y$, with probability $99\%$, it outputs $Z$ satisfying $\left| Z - \sum_{x \in X} \|x - y\|_2 \right| \le \alpha \sum_{x \in X} \|x - y\|_2 + \tilde{O}\left(\frac{R}{\alpha^{1.5}\varepsilon}\right)$.*

*Proof.* We sketch the argument since it is follows from a combination of the prior listed results. We just apply the embedding result of Theorem D.1 as well as the guarantees of our $\ell_1$ distance query data structure from Theorem B.3. The only thing left to check is the bounds of the individual coordinates after we apply the embedding. Note that with high probability, every coordinate after applying $T$ will be bounded by $\tilde{O}(R\alpha^2)$.[4] The bound follows by just plugging into the statement of Theorem B.3. □

Note that minor modifications to the one dimensional $\ell_1$ algorithm also implies the following:

**Corollary D.3.** *Let $p \ge 1$ and suppose all points in our one-dimensional datasets are in the interval $[0, R]$. Let $Z$ be the output of Algorithm 3 but with $\alpha$ scaled down by a factor of $p$ and all counts weighted instead by $(R/(1 + \alpha/p)^j)^p$ in Lines 8 and 13 of Algorithm 3. Let $Z' = \sum_{x \in X} |y - x|^p$ be the true answer. We have $\mathbb{E} |Z - Z'| \le \alpha Z' + \tilde{O}\left(\frac{R^p \log(1/p)}{\varepsilon\sqrt{\alpha}}\right)$.*

The higher dimensional version also follows in a similar manner to Theorem B.3 due to the decomposibility of $\ell_p^p$: $\sum_{x \in X} \|x - y\|_p^p = \sum_{j=1}^d \sum_{x \in X} |x(j) - y(j)|^p$.

**Corollary D.4.** *Let $Z$ be the output of Algorithm 4 but with modifications made as in Corollary D.3. Let $Z' = \sum_{x \in X} \|y - x\|_p^p$ be the true answer. We have, $\mathbb{E} |Z - Z'| \le \alpha Z' + \tilde{O}\left(\frac{R^p d^{1.5}\log(1/p)}{\varepsilon\sqrt{\alpha}}\right)$. Furthermore, Algorithm 4 is $\varepsilon$-DP. Similarly to Theorem B.4, we can get an $(\varepsilon, \delta)$-DP algorithm satisfying $\mathbb{E} |Z - Z'| \le \alpha Z' + \tilde{O}\left(\frac{R^p d \sqrt{\log(p/\delta)}}{\varepsilon\sqrt{\alpha}}\right)$.*

## D.1 AN ALTERNATE ALGORITHM FOR $\ell_2^2$

We give an alternate, simpler algorithm, with slightly better guarantees than our general $\ell_p^p$ result.

**Corollary D.5.** *There exists an $\varepsilon$-DP algorithm which answers the $\ell_2^2$ distance query with additive error $O\left(\frac{R^2 d}{\varepsilon}\right)$ in expectation and requires $O(d)$ query time.*

*Proof.* The following identity holds:

$$\sum_{x \in X} \|x - y\|_2^2 = \sum_{x \in X} \|x - \mathbb{E} X\|_2^2 + n\|y - \mathbb{E} X\|_2^2$$

where $\mathbb{E} X = \frac{1}{n}\sum_{x \in X} x$. Note that the first quantity $\sum_{x \in X} \|x - \mathbb{E} X\|_2^2$ is a scalar which does not depend on the query $y$. Thus, an alternate $\varepsilon$-DP algorithm in the $\ell_2^2$ case is to first release a (noisy) version of $\sum_{x \in X} \|x - \mathbb{E} X\|_2^2$ as well as a noisy $\mathbb{E} X$.

If all coordinates are in $[0, R]$, then changing one data point can change every coordinate of $\mathbb{E} X$ by a $R/n$ factor. Analyzing $\sum_{x \in X} \|x - \mathbb{E} X\|_2^2$ is a bit trickier since changing one data point changes

---

[4]We need to clip the the coordinates of the embedding output so that every coordinate lies in the correct range. But this event happens with high probability, so it does not affect the distance-preserving guarantees.

a term in the sum as well as $\mathbb{E}\,X$. Let $z$ denote the new mean after changing one data point in $X$ and let $\mathbb{E}\,X$ denote the old mean. We have

$$\sum_{x \in X} \|x - z\|_2^2 = \sum_{x \in X} \|x - \mathbb{E}\,X + \mathbb{E}\,X - z\|_2^2$$

$$= \sum_{x \in X} \left( \|x - \mathbb{E}\,X\|_2^2 + 2\langle x - \mathbb{E}\,X, \mathbb{E}\,X - z \rangle + \|\mathbb{E}\,X - z\|_2^2 \right).$$

Now $n\|\mathbb{E}\,X - z\|_2^2 \leq O(R^2 d/n)$ and

$$\sum_{x \in X} 2|\langle x - \mathbb{E}\,X, \mathbb{E}\,X - z \rangle| \leq 2 \sum_{x \in X} \|x - \mathbb{E}\,X\|_2 \cdot \|z - \mathbb{E}\,X\|_2 \leq O(R^2 d).$$

Thus, the simple Laplacian mechanism of adding $\text{Laplace}(O(R^2 d/\varepsilon))$ and releasing the value of $\sum_{x \in X} \|x - \mathbb{E}\,X\|_2^2$ ensures $\varepsilon/2$-DP. Then we can release the vector $\mathbb{E}\,X$ by adding $\text{Laplace}(O(Rd/(n\varepsilon)))$ noise to every coordinate, to also ensure $\varepsilon/2$-DP. Overall, the algorithm is $\varepsilon$-DP. To analyze the error, note that we get additive error $O(R^2 d/\varepsilon)$ from the noisy value $\sum_{x \in X} \|x - \mathbb{E}\,X\|_2^2$. Assuming $n$ is sufficiently large, we can easily repeat a calculation similar to above which shows that the overall additive error is at most $O(R^2 d/\varepsilon)$ in expectation. Indeed, letting $z$ denote the noisy mean we output, we have

$$\left| \|y - z\|_2^2 - \|y - \mathbb{E}\,X\|_2^2 \right| \leq 2\|y - z\|_2 \cdot \|z - \mathbb{E}\,X\|_2 + \|z - \mathbb{E}\,X\|_2^2,$$

from which the conclusion follows. □

# E    IMPROVED BOUNDS FOR THE EXPONENTIAL AND GAUSSIAN KERNELS

In this section we provide our bounds for the exponential and Gaussian kernels, improving the query time of the result of Wagner et al. (2023) stated in Theorem 2.3.

**Theorem E.1.** *Let $\alpha \in (0,1)$ and suppose $n \geq O\left(\frac{1}{\alpha\varepsilon^2}\right)$. For $h(x,y) = \|x - y\|_2$ and $\|x - y\|_2^2$, there exists an algorithm which outputs an $\varepsilon$-DP data structure for the kernel $f(x,y) = e^{-h(x,y)}$ with the following properties:*

1. *The expected additive error is $\alpha$,*

2. *The query time is $\tilde{O}\left(d + \frac{1}{\alpha^4}\right)$,*

3. *The construction time is $\tilde{O}\left(nd + \frac{n}{\alpha^4}\right)$,*

4. *and the space usage is $\tilde{O}\left(d + \frac{1}{\alpha^4}\right)$.*

Note that our bound improves upon Wagner et al. (2023) in the large dimension regime $d \gg 1/\alpha^2$, by disentangling the factors of $d$ and $1/\alpha$. We prove this via a general dimensionality reduction result, which maybe of general interest. Note that our dimensionality reduction result also implies improved bounds for KDE queries in the non-private setting as well, as elaborated in Section G.

## E.1    DIMENSIONALITY REDUCTION FOR GAUSSIAN KDE

We obtain general dimensionality reduction results for the Gaussian and exponential KDE, using variants of the Johnson-Lindenstrauss (JL) transforms. See 2 for an overview and motivations.

**Theorem E.2** (Dim. Reduction for Gaussian and exponential kernels). *Let $G : \mathbb{R}^d \to \mathbb{R}^{O(\log(1/\alpha)/\alpha^2)}$ be the standard Gaussian JL projection where $\alpha < 1$ is a sufficiently small constant. Fix a query $y \in \mathbb{R}^d$. Let*

$$z = \frac{1}{|X|} \sum_{x \in X} f(x, y),$$

$$\hat{z} = \frac{1}{|X|} \sum_{x \in X} f(Gx, Gy)$$

*for $f(x,y) = e^{-\|x-y\|_2}$ or $f(x,y) = e^{-\|x-y\|_2^2}$. Then, $\mathbb{E}\,|z - \hat{z}| \leq \alpha$.*

As stated, Theorem E.2 requires a projection matrix of dense Gaussian random variables, making the projection time $\tilde{O}(d/\alpha^2)$. We can speed this up by using the *fast* JL transform of Ailon & Chazelle (2009), which only requires $\tilde{O}(d + 1/\alpha^2)$ time, a significant speedup in the case where the original dimension $d$ is large.

**Corollary E.3.** *The same guarantees as in Theorem E.2 holds if we use the fast JL transform and project to $O(\log(1/\alpha)^2/\alpha^2)$ dimensions.*

In the proof of Theorem E.2, we use the following facts about a standard Johnson-Lindenstrauss (JL) projection using Gaussians:

**Lemma E.4** (Indyk & Naor (2007); Narayanan et al. (2021))**.** *Let $x$ be a unit vector in $\mathbb{R}^d$ and let $G$ be an (appropriately scaled) Gaussian random projection to $k$ dimensions. Then for $t > 0$:*

$$\Pr(|\|Gx\| - 1| \geq t) \leq e^{-t^2 k/8},$$

*and*

$$\Pr(\|Gx\| \leq 1/t) \leq \left(\frac{3}{t}\right)^k.$$

*Proof of Theorem E.2* . We give the full proof for $f(x, y) = e^{-\|x-y\|_2}$. Carrying out the identical steps with very minor modifications also implies the same statement for $f(x, y) = e^{-\|x-y\|_2^2}$, whose details are omitted. Fix a $x \in X$. We calculate $\mathbb{E} |f(x, y) - f(Gx, Gy)|$ (note the randomness is over $G$). We consider some cases, depending on the value of $f(x, y)$.

**Case 1:** $f(x, y) \leq \alpha$. In this case, if $\|Gx - Gy\|_2 \geq \|x - y\|_2$, then $f(Gx, Gy) \leq \alpha$, so the additive error $|f(x, y) - f(Gx, Gy)| \leq \alpha$. Thus, the only relevant event is if the distance shrinks, i.e., $\|Gx - Gy\|_2 \leq \|x - y\|_2$. If $f(Gx, Gy) \leq 3\alpha$ after the projection, then the additive error $|f(x, y) - f(Gx, Gy)| \leq O(\alpha)$. Thus, we just have to consider the event $f(Gx, Gy) > 3\alpha$.

For this to happen, we note that $\|x - y\|_2 \geq \log(1/\alpha)$, but $\|Gx - Gy\|_2 \leq \log(\alpha^{-1}/3)$. Thus, the distance has shrunk by a factor of

$$\frac{\|x - y\|_2}{\|Gx - Gy\|_2} \geq \frac{\log \alpha^{-1}}{\log(\alpha^{-1}/3)} = \frac{\log(3) + \log(\alpha^{-1}/3)}{\log(\alpha^{-1}/3)} = 1 + \frac{\log(3)}{\log(\alpha^{-1}/3)}.$$

By setting $k = O(\log(1/\alpha)^3)$ and $t = O(1/\log(1/\alpha))$ in Lemma E.4, the probability of this event is at most $\alpha$, meaning the expected additive error $\mathbb{E} |f(x, y) - f(Gx, Gy)|$ can also be bounded by $\alpha$.

**Case 2:** $f(x, y) > \alpha$. This is a more involved case, as we need to handle both the sub-cases where the distance increases and decreases. Let $f(x, y) = r > \alpha$.

**Sub-case 1:** In this sub-case, we bound the probability that $f(Gx, Gy) \leq r - \alpha/2$. The original distance is equal to $\log(1/r)$ and the new distance is at least $\log((r - \alpha/2)^{-1})$. The ratio of the new and old distances is $g(r) = \log(r - \alpha/2)/\log(r)$. Writing $r = w\alpha/2$ for $w \geq 2$, we have

$$g(r) = \frac{\log((w-1)\alpha/2)}{\log(w\alpha/2)} = \frac{\log((w-1)/w \cdot w\alpha/2)}{\log(w\alpha/2)} = 1 + \frac{\log(1 - 1/w)}{\log(w\alpha/2)}.$$

As $|\log(1 - 1/w)| = \Theta(1/w)$ for $w \geq 2$, it suffices to upper bound $|w \log(w\alpha/2)|$ in the interval $2 \leq w \leq 2/\alpha$. One can check that the upper bound occurs for $w = 2/(e\alpha)$, resulting in $\log(1 - 1/w)/\log(w\alpha/2) = \Omega(\alpha)$. Thus by taking $k = O(\log(1/\alpha)/\alpha^2)$ in Lemma E.4, the probability that $f(Gx, Gy) \leq r - \alpha/2$ is at most $\alpha$.

**Sub-case 2:** In this sub-case, we bound the probability that $f(Gx, Gy) \geq r + \alpha/2$. Again the ratio of the old and new distances is at least $\log(r)/\log(r + \alpha/2)$. Writing $r = w\alpha/2$ for $w \geq 2$, we have

$$\frac{\log(r)}{\log(r + \alpha/2)} = \frac{\log(w\alpha/2)}{\log((w+1)\alpha/2)} = 1 + \frac{\log(1 - 1/(w+1))}{\log((w+1)\alpha/2)}.$$

Thus a similar calculation as above implies that the probability of $f(Gx, Gy) \geq r + \alpha/2$ is at most $\alpha$ by setting $k = O(\log(1/\alpha)/\alpha^2)$ in Lemma E.4.

Altogether, we have bounded the probability of $|f(Gx, Gy) - f(x, y)| \geq \alpha/2$ by at most $\alpha$, meaning $\mathbb{E} |f(Gx, Gy) - G(x, y)| \leq \alpha$, as desired.

Then by linearity and the triangle inequality, it follows that

$$\mathbb{E} |z - \hat{z}| \leq \frac{1}{|X|} \sum_x \mathbb{E} |f(x, y) - f(Gx, Gy)| \leq \frac{1}{|X|} \sum_x O(\alpha) \leq O(\alpha),$$

as desired. □

We now prove Corollary E.3 where we use the fast JL transform of Ailon & Chazelle (2009). However, the fast JL transform, denoted as $\Pi$, does not exactly satisfy the concentration bounds of Lemma E.4. In fact, only slightly weaker analogous concentration results are known. Nevertheless, they suffice for our purposes. We quickly review the concentration inequalities known for the fast JL transform and sketch how the proof of Theorem F.2 can be adapted.

**Theorem E.5** (Makarychev et al. (2019)). *Let* $\Pi : \mathbb{R}^d \to \mathbb{R}^m$ *be the fast JL map of Ailon & Chazelle (2009). Then for every unit vector* $x \in \mathbb{R}^d$, *we have:*

1. *If* $t \leq \frac{\log m}{\sqrt{m}}$, *then*

$$\Pr(|\|\Pi x\|_2^2 - 1| \geq t) \leq e^{-\Omega(\frac{t^2 m}{\log m})}.$$

2. *If* $\frac{\log m}{\sqrt{m}} \leq t \leq 1$, *then*

$$\Pr(|\|\Pi x\|_2^2 - 1| \geq t) \leq e^{-\Omega(t\sqrt{m})}.$$

3. *If* $t \geq 1$, *then*

$$\Pr(|\|\Pi x\|_2^2 - 1| \geq t) \leq e^{-\Omega(\sqrt{tm})}.$$

*Proof of Corollary E.3.* We sketch the modification needed and everything else is identical to the proof of Theorem E.2. Going through the proof, we can check that Case 2 is the only bottleneck that potentially requires a higher projection dimension than Theorem E.2. Here, we need to set $t = \alpha$ in Theorem E.5 and the first inequality there is relevant. However, due to the $\log m$ factor in the denominator, we require an additional $\log(1/\alpha)$ factor in the projection dimension to achieve the same probability of failure as in the proof of Theorem E.2. □

*Proof of Theorem E.1.* We simply apply our dimensionality reduction result of Corollary E.3 in a black-box manner in conjunction with the data structure of Theorem 2.3 from Wagner et al. (2023): First we project the datapoints to dimension $\tilde{O}(1/\alpha^2)$ and build the data structure on the projected space. We also release the fast JL projection matrix used which is oblivious of the dataset so it leaks no privacy. Finally, to compute a KDE query, we also project the query vector $y$ using the fast JL projection and query the data structure we built in the lower dimensional space. The construction time, query time, and space all follow from the guarantees of the fast JL transform Ailon & Chazelle (2009) and Theorem 2.3 from Wagner et al. (2023). □

## F NEW BOUNDS FOR SMOOTH KERNELS

In this section, we give new bounds for privately computing KDE queries for the kernels $f(x, y) = \frac{1}{1+\|x-y\|_2}$, $\frac{1}{1+\|x-y\|_2^2}$, and $\frac{1}{1+\|x-y\|_1}$. Our main result is the following.

**Theorem F.1.** *Let* $\alpha \in (0, 1)$ *and suppose* $n \geq \tilde{O}\left(\frac{1}{\alpha\varepsilon^2}\right)$. *For the kernels* $f(x, y) = \frac{1}{1+\|x-y\|_2}$ *and* $f(x, y) = \frac{1}{1+\|x-y\|_2^2}$, *there exists an algorithm which outputs an* $\varepsilon$-*DP data structure with the following properties:*

1. *The expected additive error is* $\alpha$,

2. *The query time is $\tilde{O}\left(d + \frac{1}{\alpha^4}\right)$,*

3. *The construction time is $\tilde{O}\left(nd + \frac{n}{\alpha^4}\right)$,*

4. *and the space usage is $\tilde{O}\left(d + \frac{1}{\alpha^4}\right)$.*

*For the kernel $f(x,y) = \frac{1}{1+\|x-y\|_1}$, we can obtain the following:*

1. *The expected additive error is $\alpha$,*

2. *The query time is $\tilde{O}\left(\frac{d}{\alpha^2}\right)$,*

3. *The construction time is $\tilde{O}\left(\frac{nd}{\alpha^2}\right)$,*

4. *and the space usage is $\tilde{O}\left(\frac{d}{\alpha^2}\right)$.*

The road-map for this section is described in two steps. First, we give new dimensionality reduction results for the first two kernels which obtain the stronger relative error guarantee. Then we show how to combine our dimensionality reduction result with classical function approximation theory to reduce the smooth kernel case to our prior Gaussian and exponential kernel result of Theorem E.1. These results assume a similar condition on $n$ as in our Theorem E.1 and prior works Wagner et al. (2023): $n \geq \tilde{O}\left(\frac{1}{\alpha\varepsilon^2}\right)$. We present our novel dimensionality reduction for the kernels $f(x,y) = \frac{1}{1+\|x-y\|_2}$ and $\frac{1}{1+\|x-y\|_2^2}$.

### F.1 DIMENSIONALITY REDUCTION

Our main result is the following. As before, we assume the projection is chosen independently of the dataset and query.

**Theorem F.2** (Dim. Reduction for Smooth Kernels)**.** *Let $G : \mathbb{R}^d \to \mathbb{R}^{1/\alpha^2}$ be a Gaussian JL projection where $\alpha < 1$ is a sufficiently small constant. Fix a query $y \in \mathbb{R}^d$. Let*

$$z = \frac{1}{|X|} \sum_{x \in X} f(x,y),$$

$$\hat{z} = \frac{1}{|X|} \sum_{x \in X} f(Gx, Gy).$$

*for $f(x,y) = \frac{1}{1+\|x-y\|_2}$ or $f(x,y) = \frac{1}{1+\|x-y\|_2^2}$. Then, $\mathbb{E}\,|z - \hat{z}| \leq O(\alpha z)$.*

A similar corollary as Corollary F.3 also applies to the exponential and Gaussian KDE case.

**Corollary F.3.** *The same dimensionality reduction bound, up to constant factors, holds as in Theorem F.2, if we use the fast JL transform.*

*Proof of Theorem F.2.* We give the full proof for $f(x,y) = \frac{1}{1+\|x-y\|_2}$. Carrying out the identical steps with small modifications also implies the same statement for $f(x,y) = \frac{1}{1+\|x-y\|_2^2}$, whose details are omitted. Fix a $x \in X$. We calculate $\mathbb{E}\,|f(x,y) - f(Gx, Gy)|$ (note the randomness is over $G$). First we consider the case where the distance $\|Gx - Gy\|_2$ expands. Let $\mathcal{A}_i$ be the event that

$$\frac{\|Gx - Gy\|_2 - \|x - y\|_2}{\|x - y\|_2} \in [\alpha i, \alpha(i+1))$$

for $i \geq 0$. We have

$$
\sum_{i \geq 0} \Pr[\mathcal{A}_i] \, \mathbb{E}[|f(x,y) - f(Gx,Gy)| \mid \mathcal{A}_i] \leq \sum_{i \geq 0} e^{-i^2/8} \left( \frac{1}{1 + \|x - y\|_2} - \frac{1}{1 + \|x - y\|_2(1 + \alpha(i+1))} \right)
$$

$$
= \sum_{i \geq 0} e^{-i^2/8} \frac{\|x - y\|_2}{1 + \|x - y\|_2} \cdot \frac{\alpha(i+1)}{1 + \|x - y\|_2(1 + \alpha(i+1))}
$$

$$
\leq \sum_{i \geq 0} e^{-i^2/8} \frac{\alpha(i+1)}{1 + \|x - y\|_2}
$$

$$
= \frac{\alpha}{1 + \|x - y\|_2} \sum_{i \geq 0} (i+1) e^{-i^2/8}
$$

$$
< \frac{7\alpha}{1 + \|x - y\|_2}.
$$

We now handle the cases where the distance shrinks. We further subdivide this case into sub-cases where the distance shrinks by a factor $t$ satisfying $1 \leq t \leq 6$ and the sub-case where $t \geq 6$. To handle the first sub-case, let $\mathcal{B}_i$ be the event that

$$
\frac{\|x - y\|_2}{\|Gx - Gy\|_2} \in [1 + \alpha i, 1 + \alpha(i+1))
$$

for $0 \leq i \leq 5/\alpha$. Note that

$$
\mathbb{E}[|f(x,y) - f(Gx,Gy)| \mid \mathcal{B}_i] \leq \frac{1}{1 + \frac{\|x-y\|_2}{(1+\alpha(i+1))}} - \frac{1}{1 + \|x - y\|_2}
$$

$$
= \frac{\|x - y\|_2}{\|x - y\|_2 + 1} \cdot \frac{\alpha(i+1)}{1 + \|x - y\|_2 + \alpha(i+1)}
$$

$$
\leq \frac{\alpha(i+1)}{1 + \|x - y\|_2}.
$$

Furthermore, under the event $\mathcal{B}_i$, we have that

$$
\|x - y\|_2 - \|Gx - Gy\|_2 \geq \left( 1 - \frac{1}{1 + \alpha i} \right) \|x - y\|_2 \geq \frac{\alpha i}{6} \|x - y\|_2.
$$

Thus,

$$
\sum_{0 \leq i \leq 5/\alpha} \Pr[\mathcal{B}_i] \, \mathbb{E}[|f(x,y) - f(Gx,Gy)| \mid \mathcal{B}_i] \leq \sum_{0 \leq i \leq 5/\alpha} e^{-i^2/288} \cdot \frac{\alpha(i+1)}{1 + \|x - y\|_2}
$$

$$
\leq \frac{\alpha}{1 + \|x - y\|_2} \sum_{i \geq 0} (i+1) e^{-i^2/288}
$$

$$
< \frac{160\alpha}{1 + \|x - y\|_2}.
$$

For the other sub-case, write it as the union $\cup_{i=1}^{\infty} D_i$ where $D_i$ is the event that

$$
3 \cdot 2^{i+1} \geq \frac{\|x - y\|_2}{\|Gx - Gy\|_2} \geq 3 \cdot 2^i,
$$

i.e., $\|Gx - Gy\|_2$ shrinks by a factor between $3 \cdot 2^i$ and $3 \cdot 2^{i+1}$. Lemma E.4 gives us

$$
\sum_{i \geq 1} \Pr[\mathcal{D}_i] \cdot \mathbb{E}[|f(x,y) - f(Gx,Gy)| \mid \mathcal{D}_i] \leq \sum_{i \geq 1} \left( \frac{1}{2^i} \right)^k \cdot \left( \frac{1}{1 + \|x - y\|_2/2^{i+1}} - \frac{1}{1 + \|x - y\|_2} \right)
$$

$$
\leq \sum_{i \geq 1} \left( \frac{1}{2^i} \right)^{1/\alpha^2} \cdot \frac{2^{i+1}}{1 + \|x - y\|_2}
$$

$$
\leq \frac{2}{1 + \|x - y\|_2} \cdot \sum_{i \geq 1} \left( \frac{1}{2^i} \right)^{1/\alpha^2 - 1}
$$

$$
\leq \frac{2\alpha}{1 + \|x - y\|_2}.
$$

Together, the above cases imply that

$$\mathbb{E}\,|f(x,y) - f(Gx, Gy)| \leq O(\alpha) \cdot \frac{1}{1 + \|x - y\|_2}.$$

Then by linearity and the triangle inequality, it follows that

$$\mathbb{E}\,|z - \hat{z}| \leq \frac{1}{|X|} \sum_x \mathbb{E}\,|f(x, y) - f(Gx, Gy)| \leq O(\alpha) \cdot \frac{1}{|X|} \sum_x \frac{1}{1 + \|x - y\|_2} \leq O(\alpha z),$$

as desired. $\qquad\square$

*Proof of F.3.* We outline the steps in the proof of Theorem F.2 which required concentration bounds for the standard JL transform stated in Lemma E.4, and show how they can be appropriately replaced by the guarantees of Theorem E.5. The first step is the sum bounding the contribution of the events $\mathcal{A}_i$, defined as the event where

$$\frac{\|Gx - Gy\|_2 - \|x - y\|_2}{\|x - y\|_2} \in [\alpha i, \alpha(i + 1))$$

for $i \geq 0$. Here, for some smaller values of $i$, the second condition of Theorem E.5 is appropriate and for the rest, the third event is appropriate. Since the sum $\sum_{i \geq 0} i e^{-\sqrt{i}}$ converges to a constant, this portion of the proof carries through. The same comment applies where we bound the contribution of the events $\mathcal{B}_i$. The last place we need to check is when we bound the contribution of the events $\mathcal{D}_i$. Here, the third statement of Theorem E.5 is relevant, and the calculation boils down to showing the sum $\sum_{i \geq 1} e^{-\Omega(\sqrt{3 \cdot 2^i - 1})} \cdot 2^i$ converges to a constant, which is clearly true. $\qquad\square$

## F.2 Proof of Corollary 3.2

*Proof of Corollary 3.2.* Let $f(x)$ be the approximation to $x^{-1}$ in the interval $[\alpha, 1]$ given by Theorem 3.1 for $\delta = O(\alpha)$. Now consider $g(x) = \alpha \cdot f(\alpha x)$. For any $x \in [1, 1/\alpha]$, we have

$$|g(x) - x^{-1}| = |\alpha \cdot f(\alpha x) - x^{-1}| \leq |\delta/x| \leq O(\alpha)$$

where the first equality follows from the fact that $\alpha \cdot x$ is in the interval $[\alpha, 1]$ for $x \in [1, 1/\alpha]$. Thus, $g(x)$ is an additive $O(\alpha)$ approximation to $x^{-1}$ in the interval $[1, 1/\alpha]$. Now since $g$ and $x^{-1}$ are both decreasing functions of $x$, and $x^{-1} \leq \alpha$ for $x \geq 1/\alpha$, it immediately follows that $g(x)$ is an $O(\alpha)$ additive error approximation for $x^{-1}$ for *all* $x \geq 1$ (note the constant in the $O$ notation has increased). The bounds on the coefficients of $g$ in its exponential sum representation follows from the guarantees of Theorem 3.1. $\qquad\square$

We are now able to prove Theorem F.1.

*Proof of Theorem F.1.* The claimed guarantees follow from a simple combination of the tools we have developed so far, and a black-box appeal to the result of Theorem 2.3. For the kernel $f(x, y) = \frac{1}{1 + \|x - y\|_2}$, we can first perform dimensionality reduction to $O(1/\alpha^2)$ dimensions via an oblivious fast JL projection as stated in Corollary F.3. We then use the reduction given by Theorem 3.3 to instantiate $O(\log(1/\alpha))$ copies of private exponential KDE data structure of Theorem 2.3. The same procedure works for the kernel $f(x, y) = \frac{1}{1 + \|x - y\|_2^2}$. We don't have a dimensionality reduction result for the kernel $f(x, y) = \frac{1}{1 + \|x - y\|_1}$, so we just repeat the same steps as above, except we do not perform any dimensionality reduction. The guarantees follow from the guarantees of Theorem 2.3 along with the black box reduction given in Theorem 3.3. $\qquad\square$

## G Faster Kernel Density Estimation in the Non-Private Setting

Our novel dimensionality reduction results also obtain faster query algorithms for KDE queries in the *non-private* settings as well in the high-dimensional regime where $d \gg 1/\alpha^2$ where $d$ is the original dimension and $\alpha$ is our desired additive error. Indeed, we can combine our dimensionality reduction bounds of Theorems E.2 and F.2 with any KDE data structure by first projecting to a

reduced dimension and then instantiating the KDE data structure in the projected space. Since the dimensionality reduction preserves kernel sums, we can guarantee accurate answers in the projected dimension. In particular, by combining our dimensionality reduction results (the *fast* JL versions of corollaries E.3 and F.3) with prior KDE data structures whose preprocessing and query times are listed in Table 2, the following new results easily follow for KDE queries. They give improved query times for the Gaussian, exponential, and the Cauchy kernels. For the Gaussian and exponential kernels, we project to dimension $\tilde{O}(1/\alpha^2)$ where $\alpha$ is the additive error that prior data structures incur and for the Cauchy kernel, we project to dimension $\tilde{O}(1/\varepsilon^2)$, where $1 + \varepsilon$ is the multiplicative factors that prior data structures incur.

**Theorem G.1.** *By combining with Charikar et al. (2020), for the Gaussian and exponential kernels, we obtain a data structure which gives a $(1 + \varepsilon)$ multiplicative and $\alpha$ additive error guarantee for any fixed query with $90\%$ probability with the following preprocessing and query time:*

- *Gaussian kernel:   the preprocessing time is $\tilde{O}\left(\frac{nd}{\varepsilon^2 \alpha^{0.173+o(1)}}\right)$ and query time $\tilde{O}\left(d + \frac{1}{\varepsilon^2 \alpha^{2.173+o(1)}}\right)$.*

- *Exponential kernel:   the preprocessing time is $\tilde{O}\left(\frac{nd}{\varepsilon^2 \alpha^{0.1+o(1)}}\right)$ and query time $\tilde{O}\left(d + \frac{1}{\varepsilon^2 \alpha^{2.1+o(1)}}\right)$.*

*For the kernel $\frac{1}{1+\|x-y\|_2^2}$, by combining with Backurs et al. (2018), we obtain a data structure which gives a $(1+\varepsilon)$ multiplicative error with $90\%$ probability for any fixed query with preprocessing time $\tilde{O}(nd/\varepsilon^2)$ and query time $\tilde{O}(d + \frac{1}{\varepsilon^4})$.*

Table 2: Prior non-private KDE queries. The query times depend on the dimension $d$, accuracy $\varepsilon$, and additive error $\alpha$. The parameter $\beta$ is assumed to be a constant and log factors are not shown.

| Kernel | $f(x,y)$ | Preprocessing Time | KDE Query Time | Reference |
|---|---|---|---|---|
| Gaussian | $e^{-\|x-y\|_2^2}$ | $\frac{nd}{\varepsilon^2 \alpha^{0.173+o(1)}}$ | $\frac{d}{\varepsilon^2 \alpha^{0.173+o(1)}}$ | Charikar et al. (2020) |
| Exponential | $e^{-\|x-y\|_2}$ | $\frac{nd}{\varepsilon^2 \alpha^{0.1+o(1)}}$ | $\frac{d}{\varepsilon^2 \alpha^{0.1+o(1)}}$ | Charikar et al. (2020) |
| Exponential | $e^{-\|x-y\|_2}$ | $\frac{nd}{\varepsilon^2 \alpha^{0.5}}$ | $\frac{d}{\varepsilon^2 \alpha^{0.5}}$ | Backurs et al. (2019) |
| Laplacian | $e^{-\|x-y\|_1}$ | $\frac{nd}{\varepsilon^2 \alpha^{0.5}}$ | $\frac{d}{\varepsilon^2 \alpha^{0.5}}$ | Backurs et al. (2019) |
| Rational Quadratic | $\frac{1}{(1+\|x-y\|_2^2)^\beta}$ | $\frac{nd}{\varepsilon^2}$ | $\frac{d}{\varepsilon^2}$ | Backurs et al. (2018) |

# H   ADDITIONAL EXPERIMENTAL DISCUSSION

Note a simple but important point: $\varepsilon, \delta$ are input parameters, so we cannot just output the data structure or ML model with the highest accuracy. The data structure or model we output must satisfy the given privacy guarantee. Thus, accuracy vs privacy vs runtime are non-trivial trade-offs.

**Methodology of Yu et al. (2020).**   We use the "GP" baseline from Yu et al. (2020), which trains a linear classifier with DP-SGD (Abadi et al., 2016) on top of features from SimCLR (Chen et al., 2020b). Deviating from the vanilla DP-SGD, GP uses all samples to compute gradients at every iteration (i.e., no subsampling) as it was found to perform better. In our implementation, we use the "r152_2x_sk1" SimCLR network released from Chen et al. (2020b) to extract the features of the images. When training the linear classifier, we do a grid search of the hyper-parameters (learning rate $\in [0.1, 0.05, 0.1]$, gradient clipping threshold $\in [1.0, 0.1, 0.01]$, noise multiplier $\in [100, 500, 1000]$) and take the best combination. Following the common practice (Yu et al., 2020), we ignore the privacy cost of this hyper-parameter tuning process.

**Methodology of De et al. (2022).**   De et al. (2022) pretrains the WRN-28-10 network (Zagoruyko & Komodakis, 2016) on ImageNet and fine-tunes it on CIFAR10 with DP-SGD (Abadi et al., 2016). We use their official code for the experiments. We do a grid search of the noise multiplier ($\in [9.4, 12.0, 15.8, 21.1, 25.0]$) where the first four values are used in the paper and the last value is an additional one we test. We report the best results across these values and ignore the privacy cost of this hyper-parameter tuning process.

**Our hyper-parameters.** For our method, we take the embeddings from the pre-trained "r152_3x_sk1" SimCLR network released from Chen et al. (2020b). Our embeddings were in dimensions 6144. Since we are computing the $\ell_2$ distance squared, we can apply Johnson-Lindenstrauss to reduce the dimensionality, without any privacy loss. Furthermore, we can clip the embeddings as well, which reduces the overall sensitivity of our algorithm (to reduce the $R$ dependency in Section D.1) Thus, our hyper-parameters were the projection dimensions, which we looped from 100 to 2000 and the clipping threshold, which we picked from 10 choices in $[0.001, 1]$.

**Additional Results.** In Figure 4, we also show the $\varepsilon$ vs accuracy trade-off, ignoring runtime. We plot accuracy as a function of the privacy $\varepsilon$. $\delta$ is always $10^{-5}$. We also plot the best performance of every tested method: we iterate over the hyper-parameters of all methods including both De et al. (2022) and Yu et al. (2020) using their code, and we display the best accuracy for every value of $\varepsilon$. Hyper-parameters are described above. Note the trivial accuracy is .1 via random labels. We see that for small values of $\varepsilon$, we obtain the second best results, but lag behind both prior SOTA for large $\varepsilon$ regime. The accuracy in the large $\varepsilon \geq 1$ regime are $0.87, 0.93, 0.95$ respectively for ours, Yu et al. (2020), and De et al. (2022). However, our approach has a major run-time benefit compared to these prior works, as argued in Section 4. Such a boost in runtime may justify the drop in accuracy in some applications.

Note that the main bottleneck in accuracy of our method is the quality of embeddings used. If we ignore all privacy constraints, then our average similarity based methodology obtains accuracy close to $0.87$ without accounting for privacy. This is very close to the performance obtained by our $\varepsilon = 1$ private data structure of Figure 4. Thus, we cannot hope to do better in terms of accuracy. However, our method is extremely flexible in the sense that better initial embeddings, for example from other models or models pre-trained on additional or different data, can automatically lead to better downstream performance.

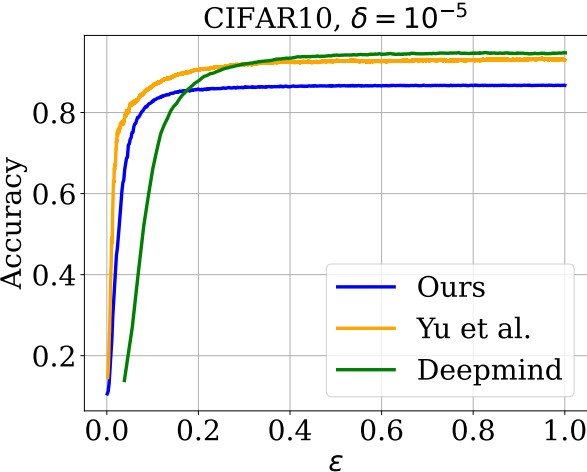

Figure 4: Accuracy of all methods as a function of $\varepsilon$, ignoring all run-time constraints. The best hyper-parameter choices are used for all methods.

# I FUTURE DIRECTIONS

We give improved theoretical algorithms for computing similarity to private datasets for a wide range of functions $f$. Our algorithms have the added benefit of being practical to implement. We view our paper as the tip of the iceberg in understanding similarity computations on private datasets. Many exciting open directions remain such as obtaining improved upper bounds or showing lower bounds for the $f$'s we considered. It is also an interesting direction to derive algorithms for more complicated 'similarity' measures, such as Optimal Transport (OT), although it is not clear what notion of privacy we should use for such measures. Lastly, generalizing our proof-of-concept experiment on DP image classification to text or other domains, using embeddings computed from models such as LLMs, is also an interesting empirical direction.

