# OpenReview forum: "Efficiently Computing Similarities to Private Datasets"
_ICLR.cc/2024/Conference — ICLR 2024 poster_

### Official Review · Reviewer_PamP · 2023-10-16

**Soundness:** 3 good
**Presentation:** 4 excellent
**Contribution:** 3 good
**Rating:** 8
**Confidence:** 3

**Summary:**

This paper gives novel results on the problem of computing the similarity between a query point and private data, in the context of differential privacy.

**Strengths:**

Originality:
Work seems original.

Quality:
Writing and presentation are very good.

Clarity:
Writing is very clear.

Significance:
The problem addressed is important for differential privacy.

**Weaknesses:**

None.

**Questions:**

Page 2:
Better put the definition of differential privacy in the main body.

Page 3:
What other notions of distance would it make sense to study?

Page 4:
I do not understand the discussion before Theorem 1.4.

Page 5:
Why is it the case that Corollary 3.2 allows you to express 1/(1 + h(x, y)) in such a way?

Page 7:
Second math display:
I do not see why the last equality holds.

Page 8:
Could you please edit the caption of Figure 1 to be more detailed?

Page 9:
I do not understand the comparison in Figure 3.

---

> ### Author Response · Authors · 2023-11-14
> **Rebuttal to Reviewer PamP**
>
> > Page 3: What other notions of distance would it make sense to study?
>
> More complicated distances such as optimal transport distance is an interesting future direction. Please see the future directions section.
>
> > Page 4: I do not understand the discussion before Theorem 1.4.
>
> The discussion outlines our algorithmic strategy for smooth kernels. We obtain new private algorithms for them via a novel privacy preserving reduction to the other kernels for which we already know how to construct private data structures, such as Gaussians. For more details, please see our Technical Overview section (section 2).
> The second part of the discussion states that our tools have additional advantages beyond privacy. They imply improved runtimes for kernel density estimation even in non-private settings.
>
> > Page 5: Why is it the case that Corollary 3.2 allows you to express 1/(1 + h(x, y)) in such a way?
>
> Please see the proof of Theorem 3.3. At a high level, we can express the function $1/(1+z)$ as a sum of a small number of exponential functions $1/(1+z) \approx \sum_i \exp(-w_i z)$ for explicit weights $w_i$. Then plugging in $h(x,y)$ for $z$ allows us to reduce KDE for the kernel $1/(1+h(x,y))$ to the KDE for $\exp(-h(x,y))$.
>
> > Page 7: Second math display: I do not see why the last equality holds.
>
> Because the $h(x,y)$ we consider are norms, such as $h(x,y) = \|x-y\|_2$, we can push any number which multiplies $h(x,y)$ outside to inside the norm expression. For example for any non-negative scalar $t$, $t \cdot \|x-y\|_2 = \| t \cdot x - t \cdot y\|_2$.
>
> > Page 8: Could you please edit the caption of Figure 1 to be more detailed?
>
> Thank you. We have updated the caption of the figure.
>
> > Page 9: I do not understand the comparison in Figure 3.
>
> Figure 3 simply highlights that a DP classification pipeline based on our private distance queries can obtain $\approx$ 85% accuracy using three orders of magnitude less time than prior state of the art methods.

---

### Official Review · Reviewer_3sGr · 2023-11-01

**Soundness:** 3 good
**Presentation:** 2 fair
**Contribution:** 2 fair
**Rating:** 8
**Confidence:** 3

**Summary:**

The authors proposed improved theoretical algorithms to compute similarity to high-dimensional private dataset for a rich class of functions. The authors give theoretical results on different distance functions and kernel functions that are advantageous over the existing results in error or query time. Further, some empirical results on $l_1$ query, dimensionality reduction and DP classification also show the advantages of the algorithm over the state-of-art benchmarks with careful discussions.

**Strengths:**

1. The authors presents solid theoretical results that improve over the existing literature on the error terms and query time for commonly used distance queries functions and KDE queries.

2. The authors presented a thorough comparisons of the proposed results with the state-of-art theoretical results in the literature.

3. Although the paper is technical, the authors provided sufficient empirical results to support the theory.

**Weaknesses:**

The paper structure can be improved. I find it more helpful to include some important formal results and algorithms in the main paper.

**Questions:**

None

---

> ### Author Response · Authors · 2023-11-14
> **Rebuttal to Reviewer 3sGr**
>
> > The paper structure can be improved. I find it more helpful to include some important formal results and algorithms in the main paper.
>
> Thank you for the feedback. We agree many important formal statements were deferred to the appendix due to the space limitations. We will bring as many of them as possible back to the main text, given more space in the final version. We have included informal theorem statements in the main body along with Table 1 which includes a summary of all our theorems and pointers to the corresponding formal statements.

---

> ### Comment · Reviewer_3sGr · 2023-11-21
>
> Thank the authors for the response. I believe adding formal results to the main paper will significantly improve the readability of the paper. I will maintain my current score.

---

### Official Review · Reviewer_8tNP · 2023-11-02

**Soundness:** 3 good
**Presentation:** 3 good
**Contribution:** 2 fair
**Rating:** 6
**Confidence:** 3

**Summary:**

This paper provides results for computing similarity-functions for queries and datasets under the constraint of differential privacy (DP). The approach they rely on mainly is leveraging the low-dimensional structures behind specific functions, whilst utilising tools, like dimensionality reduction, approximation theory, and one-dimensional decomposition of functions. They provide both theoretical and empirical results on their findings, and show that they improve on the state-of-the-art solutions for these problems.

Their algorithm for the $\ell_1$ distance function involves reduction to a series of one-dimensional decompositions. They work with Kernel Density Estimation (KDE) queries (for kernels, such as Gaussian, Exponential, and Cauchy), they use their new dimensionality reduction results that adapt the well-known JL matrices for their use-case. For the case of smooth kernels, they use functional approximation theory.

Some of their experiments are performed on the CIFAR-10 dataset.

**Strengths:**

1. The clear improvement on the prior work by Huang and Roth (2014) in terms of runtime and accuracy, both theoretically and empirically, is something to note.
2. Besides improving on prior work, this paper also has results for new kinds of queries, as well, for example $\ell_p$ distance queries.
3. The paper is easy to follow.

**Weaknesses:**

1. There is a $\sqrt{d}$ gap in the additive error for $\ell_1$ functions, and there isn't much intuition on why that may be happening or why it may be non-trivial to remove.
2. For the $\ell_1$ distance queries, the error for $\varepsilon=1$ seems very large. It will always be the case that the error decreases with increasing $\varepsilon$, but in any case, it seems like this work may not be as useful in the high-privacy regimes.

**Questions:**

1. Is there any specific reason, why you considered pure DP only, but not zCDP or approximate DP?
2. Might be worth citing this work https://arxiv.org/abs/2106.00001, since it is recent and is relevant to DP dimensionality reduction.

---

> ### Author Response · Authors · 2023-11-14
> **Rebuttal to Reviewer 8tNP**
>
> > There is a sqrt(d) gap in the additive error for L1 functions, and there isn't much intuition on why that may be happening or why it may be non-trivial to remove.
>
> We have a separate one-dimensional data structure for each of the $d$ dimensions. For pure DP, this leads to some additional overhead factor in dimension due to composition when we combine the one-dimensional data structures for the full d-dimensional data structure. Any approach exploiting one-dimensional decompositions would likely have to pay some overhead for composition. Thus a completely new approach may be required. We note that the gap can be removed at the cost of approximate-DP. This is formally stated in Theorem B.4.
>
> We remark that the naive way to perform composition leads to an $O(d)$ factor overhead for pure DP. However, we get an improved $O(\sqrt{d})$ factor by exploiting the fact that the noise used in each dimension is independent; see the proof of Theorem B.3.
>
> > For the L1 distance queries, the error for eps=1 seems very large. It will always be the case that the error decreases with increasing eps, but in any case, it seems like this work may not be as useful in the high-privacy regimes.
>
> Values such as $\epsilon \ge 3$ are widely used in real world settings [1,2,3,4]. In this case, we already achieve < 0.1 relative error whereas prior SOTA does not give any meaningful empirical result (for *any* values of $\epsilon$ tested). Theoretically in the 1-dimensional setting which our L1 experiments fall in, we have a tight matching lower bound stating that *any* DP algorithm *must* incur the same asymptotic additive error as our upper bound (see Theorem 1.2). Thus we find it unlikely that there exists an algorithm with significantly better empirical performance.
>
> We would also like to highlight our other experimental results besides L1. For example, for private classification we can obtain 85% accuracy on a standard benchmark, whereas prior state of the art methods require **$> 10^3$** times more training time to obtain the same accuracy.
>
> [1] https://blog.research.google/2023/05/making-ml-models-differentially-private.html
>
> [2] https://www.ncsl.org/technology-and-communication/differential-privacy-for-census-data-explained
>
> [3] https://docs-assets.developer.apple.com/ml-research/papers/learning-with-privacy-at-scale.pdf
>
> [4] https://desfontain.es/privacy/real-world-differential-privacy.html
>
>
> > Is there any specific reason, why you considered pure DP only, but not zCDP or approximate DP?
>
> We used pure DP since it is a harder theoretical requirement than approximate DP. All of our results easily translate to the approximate DP setting to give better results. For example, Theorem B.4 in the submission states our approximate DP result for L1 queries which removes the $\sqrt{d}$ factor. One can easily obtain similar improved results in the approximate DP setting for other queries by using Laplace noise in the places we have used Gaussian noise.
>
> > Might be worth citing this work https://arxiv.org/abs/2106.00001.
>
> Thank you for the reference. We have added it in the paper. We would like to point out that our novel dimensionality reduction results also imply faster KDE over prior SOTA even in the non-private setting; see Section G in the appendix.

---

> > ### Author Response · Authors · 2023-11-19
> > **Follow up to Reviewer 8tNP**
> >
> > Dear Reviewer 8tNP,
> >
> > Did we address all your concerns satisfactorily? If your concerns have not been resolved, could you please let us know which concerns were not sufficiently addressed so that we have a chance to respond before the November 22 deadline?
> >
> > Many thanks,
> > The authors

---

> > > ### Comment · Reviewer_8tNP · 2023-11-20
> > > **No further clarifications necessary**
> > >
> > > Thanks a lot for the detailed response! Will stick to my evaluation.

---

### Official Review · Reviewer_8sJu · 2023-11-07

**Soundness:** 3 good
**Presentation:** 3 good
**Contribution:** 2 fair
**Rating:** 8
**Confidence:** 2

**Summary:**

In this paper, the authors consider the following question: Given a private dataset X in d-dimensional space and a similarity function f(a,b) (where a and b are d-dimensional points), output a private data structure that given any y approximates the $\sum_{x\in X} f(x,y)$. In particular, the authors consider f to be distance functions such as $\ell_p$ for $p = 1, 2,\ldots,$  and also Kernel density estimates such as gaussian, exponential and Cauchy kernels. The approximation obtained is $\alpha$-relative and also has an additive factor.

In comparison to prior work, for the case where f is the $\ell_p$ norm, the relative error produced in this paper seems higher (from a relative approximation standpoint) but lower in terms of the additive error. For the KDE queries, the errors are similar to before but the parameters d and \alpha in the query time of this result are decoupled. Finally, at least some of the results in this paper seems relatively simple leading to an implementation.

**Strengths:**

The paper provides a simpler algorithm with a slightly different (better in some settings) trade-off for privately approximating distances and KDEs. The problem is an important one with many usecases. For this reason, I think the result is interesting

The techniques used involve decomposing the distance calculation into approximating several one-dimensional approximations (at least for the $\ell_1$). The idea is simple and leads to implementations.

**Weaknesses:**

The result is an improvement over existing work only under certain settings. In this sense, the result may be considered incremental. Writing is good for the first two sections, but technical insights (especially pertaining to privacy) in the main paper is somewhat limited.

Post-rebuttal update: Thank you for addressing my concerns. I have raised my score accordingly.

**Questions:**

NA

---

> ### Author Response · Authors · 2023-11-14
> **Rebuttal to Reviewer 8sJu**
>
> >  the relative error produced in this paper seems higher (from a relative approximation standpoint) but lower in terms of the additive error.
>
> > The result is an improvement over existing work only under certain settings.
>
> We can easily translate our relative and additive error to only additive error which always improves over prior works. Our bounds allow us to trade off relative error and additive error by selecting $\alpha$ (we can use any choice of $\alpha$). If we let $\alpha = d^{1/3}/(\epsilon n)^{2/3}$ and use the fact that the *largest* possible L1 query value is at most $O(nd)$, our algorithm gives additive error $\alpha \cdot O(nd) + \frac{d^{1.5}}{\epsilon \sqrt{\alpha}} = O\left( \frac{d^{4/3}n^{1/3}}{\epsilon^{2/3}}\right)$ **which is always asymptotically better** than Huang et al (they have a larger dimension factor of $d^{7/3}$). Furthermore, our experiments demonstrate that our algorithm gives much better empirical results than prior work. See Figure 2.  A similar statement holds for $\ell_2$ queries. By parameterizing our final bounds using relative error, we can get even better guarantees than prior work when query values are small.
>
> Note that if the query values are smaller, which is natural, we get an even better additive error with no relative error.
>
> Lastly, we remark that for $p > 2$, we obtain the first result for $\ell_p^p$ queries as well as the first results for many smooth kernels such as $\frac{1}{1 + \|x-y\|_2}$. There are no prior works to compare against for these new results.
>
>
> > In this sense, the result may be considered incremental.
>
> We respectfully disagree with this assessment. Besides the improved additive error as discussed above, we introduce completely new tools in the design of private algorithms for computing similarities. Our tools have implications beyond private algorithms. For example, our dimensionality reduction results imply faster KDE queries even in the non-private setting.
>
>
> > technical insights (especially pertaining to privacy) in the main paper is somewhat limited.
>
> Thank you for your feedback. For the final version, given more space, we will move additional technical content from the appendix.
>  At the same time, we refer the reviewer to Section 2 of the main body which outlines how the techniques interact with privacy. For example, as stated in Section 2, our function approximation result crucially exploits the fact that smooth kernels can be written as a *small* number of exponential sums. This allows us to reduce private KDE for the kernel $\frac{1}{1 + \|x-y\|_2}$ to private KDE for the kernel $e^{-\|x-y\|_2}$ without a blowup in the privacy parameters. We can then use existing private data structures for the kernel  $e^{-\|x-y\|_2}$. This reduction also works for other smooth kernels and is formalized in Theorem 3.3 in the main body.

---

> > ### Author Response · Authors · 2023-11-19
> > **Follow up to Reviewer 8sJu**
> >
> > Dear Reviewer 8sJu,
> >
> > Did we address all your concerns satisfactorily? If your concerns have not been resolved, could you please let us know which concerns were not sufficiently addressed so that we have a chance to respond before the November 22 deadline?
> >
> > Many thanks,
> > The authors

---

> > > ### Author Response · Authors · 2023-11-22
> > > **Second follow up to Reviewer 8sJu**
> > >
> > > Dear Reviewer 8sJu,
> > >
> > > We believe we have addressed your main misunderstanding of our work and shown that our bounds are always better in all settings. Since the discussion period is drawing to a close, we would greatly appreciate your engagement.
> > >
> > > Many thanks, The authors

---

> > > > ### Comment · Reviewer_8sJu · 2023-11-22
> > > > **thank you**
> > > >
> > > > Thank you for your response. My concerns have been addressed. I plan to read the paper again (in light of your additional inputs) before I update my score.

---

> > > > > ### Author Response · Authors · 2023-11-23
> > > > > **Thank you**
> > > > >
> > > > > Thank you, please let us know if we can be of further assistance in clarifying any points.

---

### Author Response · Authors · 2023-11-14
**Response to reviewers**

We thank the reviewers for their valuable feedback. Answers are given in a response to each review.

---

### Meta-Review · Area_Chair_DgNv · 2023-12-05

**Metareview:**

The paper studies the problem where given a similarity function f and a large high dimensional dataset, one would like to output a differentially private data structure that approximates the sum of all f(x, y) for all values of x, for any given query y. The paper considers the special cases where f is a kernel function or a distance function. On the theoretical side, the paper improves on the results of Huang-Roth (SODA 2014). The paper also shows that the proposed algorithms improve on the state-of-the-art in terms of query times and accuracy. The paper will be a nice addition to the program at ICLR.

**Justification For Why Not Higher Score:**

There is still a sizable gap (of ~ sqrt(d)) to optimality that the paper does not close.

**Justification For Why Not Lower Score:**

The paper studies a natural problem and provides a nice improvement over the prior work.

---

### Decision · Program_Chairs · 2024-01-16

Accept (poster)